

# Spaceborne thermal infrared observations of Arctic sea ice leads at 30 m resolution

Yujia Qiu[1, 2, 3], Xiao-Ming Li[2, 1], Huadong Guo[2, 1]

[1]Key Laboratory of Digital Earth Science, Aerospace Information Research Institute, Chinese Academy of Sciences, Beijing
100094, China
[2]International Research Center of Big Data for Sustainable Development Goals, Beijing 100094, China
[3]University of Chinese Academy of Sciences, Beijing 100049, China

*Correspondence to*: Xiao-Ming Li (lixm@radi.ac.cn)

**Abstract.** Sea ice leads are elongated fractures within sea ice cover, playing an important role in the heat exchange from the
ocean to the overlying atmosphere. Narrow leads less than a hundred meters in width contribute considerable heat fluxes, requiring fine-scale observation of Arctic leads. With the launch of Sustainable Development Science Satellite 1 (SDGSAT-1) by China on 5 November 2021, the on-board Thermal Infrared Spectrometer (TIS) provides thermal infrared imagery at an unprecedented resolution of 30 m in a swath of 300 km. We propose a method adapted to the TIS high-resolution infrared images for lead detection in the Arctic. For the first time, the spatial resolution of leads by infrared remote sensing increases
from the scale of kilometers to tens of meters. For the Beaufort Sea cases in April 2022, the detection is consistent with the Sentinel-2 visible images, yielding an overall accuracy of 96.30%. Compared with the Moderate-Resolution Imaging Spectroradiometer (MODIS), the TIS presents more leads with width less hundreds of meters than the results based on the MODIS data. For the three infrared bands of the TIS, the B2 (10.3-11.3 μm) and B3 (11.5-12.5 μm) bands, show similar performances in detecting leads. The B1 band (8.0-10.5 μm) can be complementary to the other two bands, as the
temperature measurement sensitivity is different from the other two, benefiting better detection by combining the three bands. This study demonstrates that SDGSAT-1 TIS data at 30 m resolution is well applicable for observing previously unresolvable ice leads, and will provide insight into the contribution of narrow leads to rapid sea ice changes in the Arctic.

## 1 Introduction

Over several decades, the Arctic has warmed at approximately twice the rate as the entire globe, as the result of a well-
known phenomenon of Arctic amplification (Arrhenius, 1896) that has attracted increasing attention. Among a suite of Arctic amplification causes and processes, the ongoing changes in the Arctic sea ice extent and the heat fluxes between the ocean and the atmosphere are prominent (Serreze and Barry, 2011). Leads are elongated fractures within sea ice cover, which develop as the result of sea ice fracturing under wind and water stresses. Open water in leads may refreeze when exposed to a cold atmosphere, so leads contain unfrozen water and ice of varying thicknesses. Although the area of these
openings is relatively small, covering less than 2% of the central Arctic, leads hold significant importance for the Arctic



mass and heat balance (Vihma et al., 2014). A small change of 1% in the lead fraction would cause a large fluctuation in the air temperature by up to 3.5 K (Lüpkes et al., 2008). Leads provide windows for heat exchange between air and water, contributing to more than 70% of the upward heat flux (Marcq and Weiss, 2012). During winter, newly opened leads are the main source of ice production, brine rejection, and turbulent heat loss to the atmosphere (Maykut, 1982; Alam and Curry,

1998). In spring, surface melt creates more openings and releases more heat into the atmosphere (Ledley, 1988; Tschudi et al., 2002). As preferential melting sites in early summer (Alvarez, 2022), leads strongly absorb shortwave radiation during the melting season, promoting lateral and basal melt of sea ice (Maykut, 1982), accelerating sea ice thinning (Kwok, 2018) and decreasing the mechanical strength of sea ice (Gimbert et al., 2012); these processes enable a more considerable drifting speed, deformation, and possibly a faster export (Rampal et al., 2009; Onarheim, 2018). In turn, more fracturing and earlier

openings are expected to create more intensive networks of leads in the following spring (Steele et al., 2015).

Under the ongoing trend of sea ice retreat in the Arctic (Cavalieri and Parkinson, 2012; Stroeve et al., 2012), identifying the characteristics of sea ice leads can help enhance our understanding of thermodynamic and mechanical processes in the Arctic. Various remote sensing instruments have been used for sea ice lead research since the early 1990s, especially by moderate-resolution thermal infrared satellite images, e.g., the Advanced Very High-Resolution Radiometer (AVHRR) (Key et al.,

1993; Lindsay and Rothrock, 1995), Moderate-Resolution Imaging Spectroradiometer (MODIS) (Willmes and Heinemann, 2015a and 2015b; Hoffman et al., 2019 and 2021; Reiser et al., 2020; Qu et al., 2021), and Landsat-8 Thermal Infrared Sensor (TIRS) (Qu et al., 2019) data. Other studies also applied active and passive microwave data to lead detection with the advantage that microwave wavelengths are transparent to cloud cover; however, either the data resolution is too coarse, e.g., with Advanced Microwave Scanning Radiometer for Earth Observing System (AMSR-E) with a resolution of 6.25 km

(Röhrs and Kaleschke, 2012; Bröhan and Kaleschke, 2014) or the observations are discontinuous, e.g., by synthetic aperture radar (SAR) (Murashkin and Spreen, 2019; Liang et al., 2022) and altimeter (Wernecke and Kaleschke, 2015; Lee et al., 2018).

For sea ice lead detection based on thermal infrared data, the key lies in deriving thermal contrasts, namely, the temperature anomaly between sea ice and open water. To this end, various temperature datasets were used in previous studies. Willmes

and Heinemann (2015a) utilized the MODIS ice surface temperature (IST) product to map pan-Arctic lead distribution from January to April over the period of 2003 to 2015. The long-term daily lead product is available to assess seasonal divergence patterns of sea ice in the Arctic Ocean (Willmes and Heinemann, 2015b). Essentially, IST data, which are generally retrieved by the split-window technique (Key et al., 1997), are less accurate under the presence of melt ponds and leads because the lower emissivity (0.96 compared to 0.99) can cause a difference in the retrieved temperature (Hall et al., 2001). Furthermore,

cloud masking defects affect lead detection (Hoffman et al., 2019; Reiser et al., 2020). Hoffman et al. (2019) focused on using at-sensor brightness temperature (BT) data and improved cloud masking. They detected leads for January through April over the period of 2003 to 2018, presenting a lower estimation for the lead area compared with the results in Willmes and Heinemann (2015c); the reason is the difference in spatial resolutions of the lead datasets. The recently published work applied the convolutional neural network U-Net to Visible Infrared Imaging Radiometer Suite (VIIRS) 11 μm BT images for



lead detection (Hoffman et al., 2021). The analysis of lead area over the winter season from 2002 to 2022 had a small

decreasing trend due to increasing cloud cover in the Arctic, but an increasing trend of 3,700 km$^2$ per year after removing the

impact of cloud cover changes (Hoffman et al., 2022a). Qu et al. (2021) proposed a modified algorithm to detect daily spring

leads in the Beaufort Sea based on the IST data retrieved from MODIS swath products, providing better results in terms of

identifying open water leads and refrozen leads; they found a positive interannual trend in the April lead area for the study

period of 2001 to 2020 of approximately 2,612 km$^2$ per year.

Adequate lead observations have important contributions to understanding rapid sea ice changes in the Arctic Ocean (Zhang

et al., 2018; Ólason et al., 2021). Narrow leads of less than a hundred meters in width are over two times more efficient at

transmitting turbulent heat than larger leads of hundreds of meters (Marcq and Weiss, 2012). However, due to the limitations

of spaceborne thermal infrared sensors in terms of spatial resolution, current lead observations are available only up to

moderate resolution on a kilometer scale. The narrowest lead widths that can be revealed by the detection are several

kilometers or even coarser (Key et al., 1993). Qu et al. (2019) resampled Landsat-8 TIRS data with a resolution of 100 m to

30 m to estimate heat fluxes over the detected leads. Their result showed an underestimated lead information detected by

MODIS data compared to TIRS data, owing to the inability of MODIS to resolve small leads (widths smaller than 1 km).

Consequently, the heat flux estimation from Landsat-8 TIRS data is larger than that from MODIS data, where small leads

contribute to more than a quarter of the total heat flux. Yin et al. (2021) proposed a convolutional neural network-based

framework to estimate turbulent heat flux over leads at the sub-pixel scale based on MODIS data. The super-resolution

estimates are better than those estimated by original moderate resolution data (1 km) and by interpolation-based high

resolution data (100 m), but still have limitations for very narrow leads. Therefore, the spatial resolution on a kilometer-scale

does not support the parameterization of actual lead characteristics in the Arctic Ocean. High-resolution observations are

urgently needed to reveal narrow leads and their variability processes.

An emerging opportunity to obtain high-resolution observations is the Sustainable Development Science Satellite 1

(SDGSAT-1), which was successfully launched on November 5, 2021, and is the first satellite customized for the United

Nations (UN) 2030 Agenda for Sustainable Development (Guo et al., 2022). Three payloads, the thermal infrared

spectrometer (TIS), Glimmer Imager for Urbanization (GIU), and Multispectral Imager for Inshore (MII), allow the satellite

to obtain high-quality data as well as full-time monitoring capabilities to facilitate the evaluation of SDG indicators (Guo,

2019; Guo et al., 2022). The TIS is used for global thermal radiation detection with three thermal infrared bands (see Table 1

for the sensor characteristics). More importantly, the TIS has a spatial resolution of 30 m, parallel with a wide swath of 300

km. With such an unprecedented infrared imaging capability, SDGSAT-1 TIS is expected to provide far more details of sea

ice characteristics in polar regions than current thermal infrared sensors in orbit. To date, the TIS has acquired substantial

high-resolution thermal infrared data from the critical seas in the Arctic, e.g., the Beaufort Sea and the Laptev Sea. Figure 1

presents few cases in March and April 2022 under clear sky conditions. Under such attractive prospects, we pioneered the

scientific application based on SDGSAT-1 TIS data to examine its feasibility in detecting sea ice leads from the Arctic

Ocean. With regard to the thermal characteristics of high-resolution data, we proposed an improved lead detection method



based on a combination of a binary segmentation and a designed filter. To determine the reliability of the detailed features

resolved at 30 m resolution, a series of comparisons were performed, including comparisons with visible and SAR data at high resolutions, as well as comparisons with comparable sea ice lead products at moderate resolutions.

This study is the first to observe Arctic sea ice leads at 30 m resolution and reveal the details that are unresolvable by moderate-resolution thermal infrared sensors. The results will help to understand the processes of Arctic lead variability and its contribution to Arctic sea ice retreat. The paper is organized as follows. Section 2 introduces the data used in this study,

including SDGSAT-1 TIS data for lead detection, visible images for validation, and others for comparative analysis. Section 3 presents the method applied to derive sea ice leads. Section 4 presents the high-resolution lead detection results of this study, the validation against visible images, the cross-comparison among three infrared bands, and the comparison with moderate-resolution results. In Section 5, we explore the factors affecting lead detection and the lead properties resolved by high-resolution imagery. Finally, a summary and conclusion are given in Section 6.


**Table 1. SDGSAT-1 TIS characteristics and radiometric performance**

| | |
|---|---|
| Spatial resolution | 30 m |
| Swath width | 300 km |
| Revisit time | 11 days |
| Band wavelengths | B1: 8.0-10.5 μm |
| | B2: 10.3-11.3 μm |
| | B3: 11.5-12.5 μm |
| Dynamic range | 220 K-340 K |
| Noise equivalent differential temperature (NEΔT) | 0.2 K @300 K |
| Radiometric calibration accuracy | Absolute radiometric calibration: ≤1K, Relative radiometric calibration: 5% |



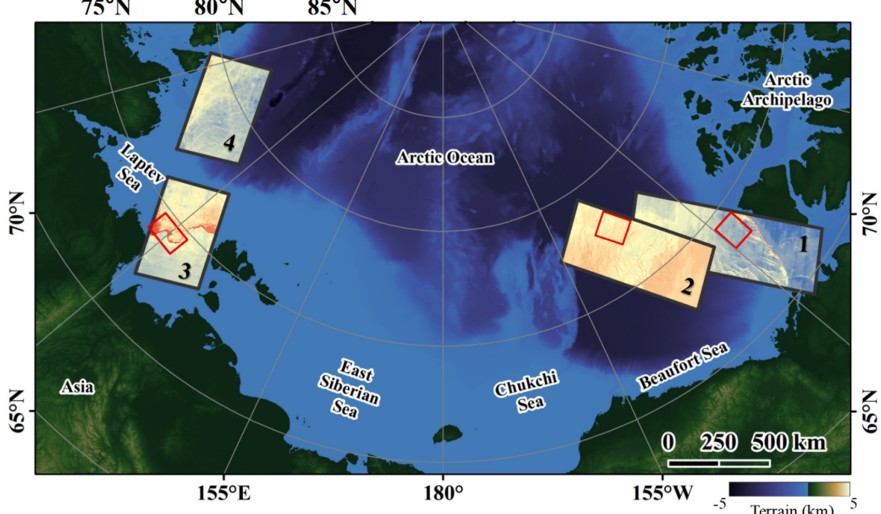

**Figure 1: Geospatial distributions of SDGSAT-1 TIS data collected from the Arctic Ocean in March and April in 2022 used in this**
**study for sea ice lead detection. The black borders mark four successive groups of cloudless images where the color represents the**
**BT values from the TIS B2 band. The small red squares are the regions where the TIS data are matched with the visible images**
**for validation.**

## 2 Data

### 2.1 SDGSAT-1 TIS

As listed in Table 1, the TIS has three infrared bands, which are centered at 9.3 μm (8.0-10.5 μm, Band 1 (B1)), 10.8 μm
(10.3-11.3 μm, Band 2 (B2)), and 11.8 μm (11.5-12.5 μm, Band 3 (B3)) and has the ability to resolve temperature
differences as low as 0.2°C (@ 300 K) (Guo et al., 2022). In the commissioning phase of the satellite, the analysis shows that
the accuracy of the radiometric measurement is better than 0.42 K for the three bands (Hu et al., 2022), which satisfies the
preflight requirements (≤1 K). The B2 and B3 bands are the two split-window channels widely used in surface temperature

retrieval, while the B1 band is not commonly used in infrared observation missions. Liu et al. (2021) estimated the ability of
SDGSAT-1 TIS data to retrieve land surface temperature when different split-window algorithms were applied, i.e., the
generalized split-window algorithm using the B2 and B3 bands and the three split-window algorithm using the B1, B2 and
B3 bands together. Their results showed that the three-band method may perform better than the two-band method with a
root mean square error lower than 1 K.

Considering the benefit of incorporating three thermal infrared bands for observation, Thus, the three bands of SDGSAT-1
TIS data are used for detection of ice leads in this study. The georeferenced level-4 TIS data in the Beaufort Sea and the



Laptev Sea in the spring season of 2022 were collected and manually selected for cloud coverage of less than 10% for sea ice lead detection. Four scenarios of the 11 TIS data is shown in Figure 1 (see Table 2 for the data information).

**2.2 Sentinel-1 and Sentinel-2**

Sentinel-2 (S2) is formed by two satellites, S2A and S2B. Both satellites are equipped with a Multispectral Instrument (MSI) with thirteen spectral channels covering the visible, near-wave and shortwave infrared spectral zones. Level-1c S2 products provide top-of-atmosphere reflectance processed in radiometric and geometric corrections in tile form. Each tile is an ortho-image in a 100 by 100 $km^2$ area. S2 MSI green band images at a resolution of 10 m are used for comparison with the leads detected by TIS for validation, given that the visible spectrum centered at 560 nm gives a good effect (König et al., 2019) for

a scene containing sea ice and seawater. Images acquired over the Beaufort Sea and the Laptev Sea in March and April 2022 were collected (see Table 2 for the data information, and see the red squares in Figure 1 for the location).

Sentinel-1 (S1) is a C-band SAR imaging day and night regardless of the weather. Both S1A and 1B acquire dual-polarization (HH and HV) imagery, covering the vast Arctic region. The S1 extra-wide (EW) swath data have a swath width of approximately 400 km, with a pixel size of 40 m by 40 m. We used the S1 level-1b data in the format of ground range

detected medium resolution (GRDM). As S1B has been out of work since December 2021, only the S1A data are available in this study period. Considering that the backscatter values of SAR in different polarizations give different sensitivities for leads fully opened or covered by thin ice, we collected S1A dual-polarization data in the Beaufort Sea on April 3 and 28, 2022 (see Table 2).

**2.3 MODIS IST product**

The MODIS is an instrument onboard the two polar-orbiting satellites Terra and Aqua, which are part of NASA's Earth Observing System (EOS). The MODIS acquirs data in 36 discrete spectral bands from the optical to the thermal infrared radiance wavelength region. The swath width of the MODIS is 2330 km. The MOD29P1D and MYD29P1D daily level-3 sea ice products include daily sea ice cover and IST datasets (Hall and Riggs, 2021). Each product contains a tile of data gridded into the Lambert azimuthal equal-area map projection, which is approximately 1200 by 1200 $km^2$ in area. The IST

datasets have a spatial resolution of 1 km. The IST data are retrieved by the split-window technique based on the MODIS 31 and 32 bands, with an accuracy of 1.2-1.3 K (Hall et al., 2004). Cloud masking from the MODIS cloud products for daytime and nighttime (Ackerman et al., 1998) is integrated into the IST retrieval. Given that the SDGSAT-1 mainly passes over the Laptev Sea and the Beaufort Sea in the morning, IST data produced by MODIS/Terra, i.e., MOD029P1D products, are mainly used in this study (see Table 2).

**2.4 ERA5 air temperature data**

European Centre for Medium-Range Weather Forecasts (ECMWF) provides the fifth-generation reanalysis data (ERA5) for global climate and weather for the past seven decades (Hersbach et al., 2018). The ERA5 near-surface air temperature (2 m



air temperature) data available by every 6 hours and in a regular grid of 0.25 degrees. In this study, we used 2 m air temperature data from March to April 2022 to analyze the possible variations in the atmospheric environment.

**2.5 OMI/Aura product**

Since the TIS B1 band covers an absorption channel for ozone (Wan and Li, 1997), we analyzed the possible absorption effects of different ozone resolutions on thermal infrared radiation in this study. The Ozone Monitoring Instrument (OMI) is an instrument onboard the EOS Aura mission. The OMI measurements cover a spectral region of 264–504 nm, which aims to continue the record for total ozone and other atmospheric parameters related to ozone chemistry and climate. The level-3
OMI/Aura Ozone Total Column data (OMTO3) are produced by using best pixel data from approximately 15 orbits, covering the whole globe and mapped in a grid size of 0.25 degrees (Bhartia, 2012).

**Table 2. Information of satellite data and derived product used in this study**

| | | SDGSAT-1 TIS | Sentinel-2 MSI | MOD029P1D | Sentinel-1A EW |
|---|---|---|---|---|---|
| | **2022-03-23** | 10:52:13<br>10:52:59<br>10:53:43<br>10:54:13 | 03:55:34 | h07/08/09/10<br>v07/08/09 | / |
| **Date and time (UTC)** | **2022-04-03** | 04:26:39<br>04:27:09<br>04:27:39<br>04:28:09 | 21:00:23 | h07/08/09/10<br>v07/08/09 | 15:53:09 |
| | **2022-04-28** | 04:56:25<br>04:55:26<br>04:55:55 | 22:42:28 | h07/08/09/10<br>v07/08/09 | / |
| **Spatial resolution** | | 30 m | 10 m | 1 km | 40 m |

**3 Method**

In this section, we propose a method for sea ice lead detection adaptable to high-resolution TIS images, based on the principle of exploiting both the relative and absolute temperature characteristics of ice leads.



### 3.1 Pre-processing of TIS data

The SDGSAT-1 TIS performs on-orbit radiometric calibration using an onboard black body. First, all digital numbers

(*DNs*) are converted into at-sensor radiance using formula (1).

$$L = gain \times DN + bias - bg \ , \tag{1}$$

where the gain and bias are radiometric calibration coefficients provided by the scientific calibration team, which have included relative and absolute radiometric calibrations; $bg$ is the background radiance of the black body. Then, the BT is calculated from the at-sensor radiance using the Planck function. In the following section, the at-sensor BT of the three

thermal infrared bands is used for sea ice lead detection.

### 3.2 Detection of ice leads

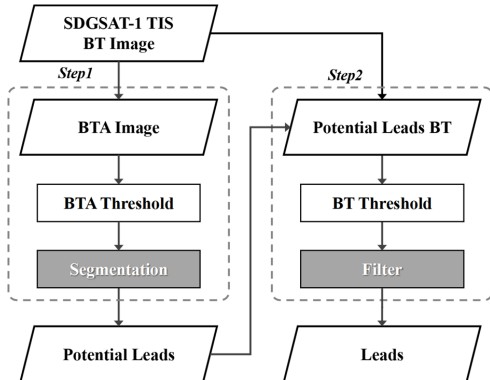

**Figure 2: Flowchart of sea ice lead detection based on SDGSAT-1 TIS data.**

Ice leads containing seawater and thin ice have temperatures higher than the surrounding sea ice. Therefore, the temperature

contrast between leads and the sea ice surface is the basis of detecting ice leads. However, as the spatial resolution of thermal infrared imagery is improved, the temperature variations in sea ice with different thicknesses pose a challenge for accurate lead identification. Figure 2 shows the flowchart for detecting leads based on SDGSAT-1 TIS data in this study, which contains a segmentation and a filter. The algorithm's input is the BT data of each TIS band (B1, B2, and B3 bands), as shown in Figure 3 (a) for an example of the B1 band. Thanks to the high spatial resolution of 30 m, the thermal features of

sea ice and leads are clearly observable. In addition to the leads presenting as distinct yellow and red colors on the BT map, slight variations in sea ice surface temperature can be identified from approximately 237 K to 242 K. The brightness temperature anomaly (BTA) images are derived from the BT data by subtracting the mean temperature in neighbour windows with sizes of 2.4 km by 2.4 km (80 pixels by 80 pixels), as shown in Figure 3 (b). Undoubtedly, the BTA data





further highlight the presence of leads, but the positive BTA values caused by thinner sea ice are also highlighted. To this

end, binary segmentation extracts potential leads from the BTA data. A designed filter is further applied to the segmentation, and the leads are consequently obtained. The following two subsections describe the two major steps involved in the proposed method.

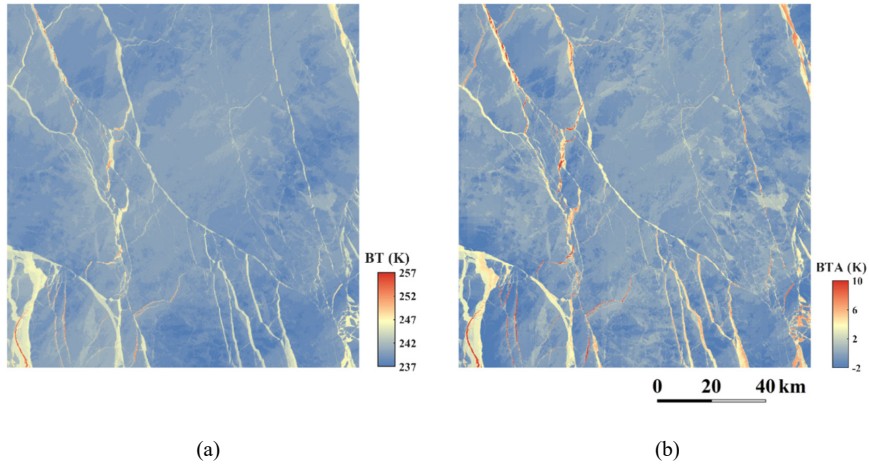

(a)                                                                                      (b)

**Figure 3: Example for the BT image and BTA image based on the SDGSAT-1 TIS B1 band (8-10.5 μm) acquired on April 3 in 2022. (a) The BT image. (b) The derived BTA image.**

**SDGSAT-1 TIS data ID: KX10_TIS_20220403_W128.84_N73.00_202200033226.**

### 3.2.1 Potential segmentation of ice leads based on the BTA data

The key to performing a binary segmentation by the BTA data is to find an appropriate threshold to segment sea ice and leads. By collecting eight TIS data acquired between April 3 and April 28, 2022 in the Arctic Ocean, we analyzed the distribution of their BTA data, as shown in Figure 4. The BTA data show a normal distribution, as the overlaid Gaussian

fitting (with μ = -0.25 K, $\sigma^2$ = 0.38 K) indicates. The peak in the histogram appears at -0.25 K, accounting for 15.09% of all the data. The long tail on the positive side of the histogram suggests that the images contain leads as they have a higher temperature than the sea ice. Thus, we need to determine a threshold in the positive BTA to segment the leads from other features.



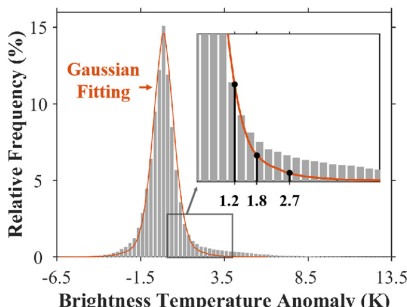

**Figure 4: Statistical BTA histogram of eight TIS data acquired from April 3 to April 28, 2022, with a bin width of 0.25 K. The orange curve is the Gaussian fitting, with μ = -0.25 K and σ² = 0.38 K.**

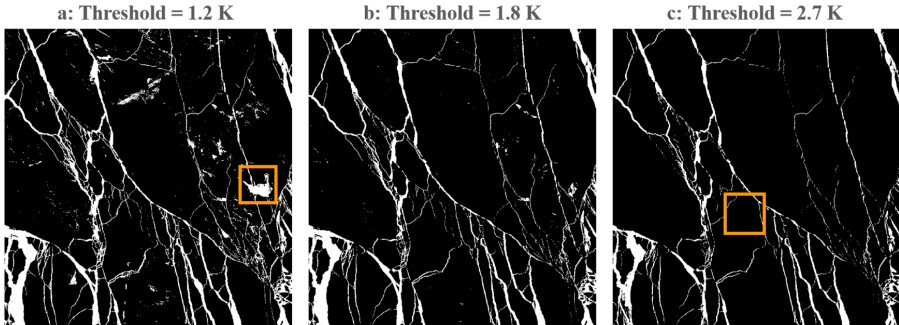

**Figure 5: BTA threshold test results for potential lead segmentation (using different thresholds of 1.2, 1.5 and 2.7 K, from left to right). Pixels with BTA greater than or equal to the threshold are classified as 1 and represented by the white areas; pixels with BTA less than the threshold are assigned a value of 0, as shown by the black background. The orange squares contain false detections.**

Previous methods applied a variety of BTA thresholds for lead detection. Based on BTA derived from the MODIS IST product, Willmes and Heinemann (2015a) compared the standard deviation and a set of non-parameterized methods. In terms of BTA derived by MODIS 11 μm swath data, Hoffman et al. (2019) identified a threshold of 1.5 K. Qu et al. (2021) took 1.2 K, 1.5 K and 2 K as thresholds for different types of leads, corresponding to large to small uncertainty levels. We enlarge part of the histogram tail in Figure 4. The Gaussian curve gradually deviates from the bars when the BTA value is greater than 1.2 K, which should indicate a transition from ice to leads. We tested various thresholds and found that choosing 1.2 K, 1.8 K, and 2.7 K as thresholds yields distinguishable differences in the segmentation results, as one example presented in Figure 5. Using a threshold of 1.2 K results in false-positive detections (i.e., sea ice classified as leads), e.g., the white pixels marked by the orange square in Figure 5 (a) (this can be identified in the original BTA map shown in Figure 3 (b)). In contrast, using 2.7 K as the threshold results in a loss of detail, e.g., the part marked by the orange square in Figure 5 (c) (compared to Figure 5 (b)). Multiple tests using 1.8 K as the BTA threshold avoids many false-positive detections while still





capturing lead details. The BTA threshold of 1.8 K was applied to all SDGSAT-1 TIS data in this study for potential lead segmentation.

**3.2.2 Further filter based on a BT threshold**

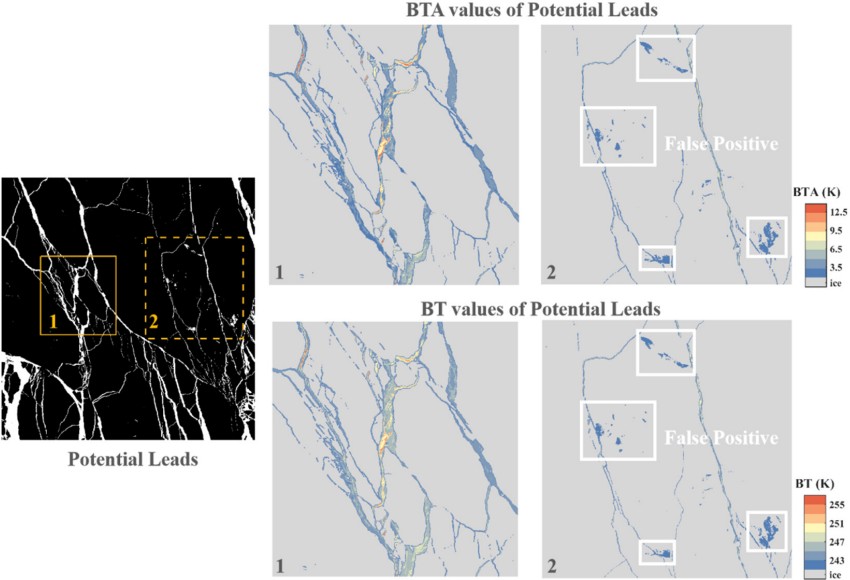

**Figure 6: Characteristics of potential leads after segmentation. The binary image on the left panel presents the leads detected by segmentation, where the two squares are 1: highly reliable detection and 2: part of false-positive detection. For these potential leads in both views, their BTA images are shown in the first row on the right panel, and the BT images are shown in the last row, 240 where the gray background represents the ice surface.**

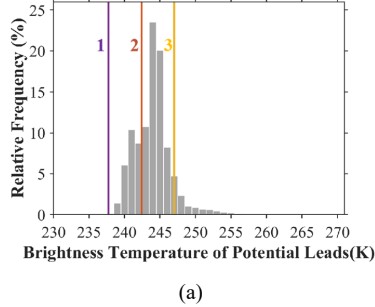

(a)



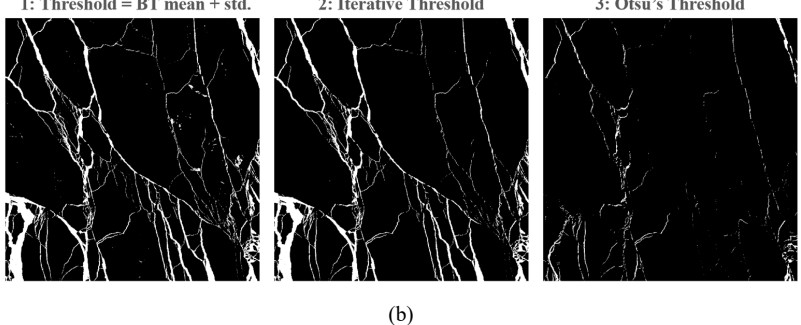

(b)

**Figure 7: BT threshold tests and filtered results based on the thresholds. (a) BT histogram of potential leads overlaid with three selections for the BT threshold of the filter. (b) The filtered results, where the pixels with BT values below the threshold are rejected and classified as background. These thresholds are 1: mean plus standard deviation (std) of BT before segmentation, 2: iterative threshold, and 3: Otsu's threshold.**

Following the segmentation conducted in the previous step, a few false-positive detections remain in the result. This situation is unavoidable to some extent because masses of information on the sea ice surface are also resolved by high-resolution thermal infrared data. Even small changes in the temperature gradient could yield high BTA values, resulting in false-positive detections. We decided to identify the reliability of those potential leads for better detection accuracy. On the

left panel of Figure 6, the potential leads in the square marked by solid yellow lines are considered reliable, while part of the white pixels marked by the other square (with dashed lines) are false-positive detections. The BTA data of the detected leads in the two views are displayed in the first row on the right panel, and the corresponding BT data are shown in the next row. Whether for the BTA or BT data, the dark blue pixels (marked by white squares) are more likely to represent those false-positive detections. However, it is difficult to evaluate further the reliability of potential leads based on only the BTA data.

In contrast, false-positive detections can be easily distinguished from leads based on BT data. For example, the absolute values of the BT of those reliable leads in the first column (in view 1) are all greater than those of the false-positive detections in the second column (in view 2) by at least 2 K.

The BT histogram for those remaining potential leads is shown in Figure 7 (a). Those pixels with low temperature on the left side represent the false-positive detections; the high-frequency pixels and the tail on the right are those highly reliable leads.

Thus, we used a filter to remove the pixels with BT values below a given threshold. Unlike using the BTA threshold as a constant, the threshold determined for the BT data is adaptive for environmental variations. In this regard, we tested non-parameterized threshold selection methods, including Otsu's threshold (Otsu, 1979), iterative selection (Ridler and Calvard, 1978), and the threshold based on the BT mean and standard deviation (calculated by the BT map before segmentation). The result in Figure 7 (b) suggests that the iterative threshold filter performs the best because it rejects false detections. The mean

and standard deviation filter ranks second. The Otsu's threshold is not adapted for use in this filter. Therefore, the iterative selection determines the BT threshold in this filter. The starting position of the iteration is set to the sum of the BT mean and





standard deviation, which can save iterative times. For each TIS band, the respective threshold was selected and the pixel with BT value above the threshold was filtered out. Finally, the binary detection of leads at a 30 m resolution was derived based on SDGSAT-1 TIS in three bands.

**4 Results**

This section presents the derived sea ice leads at a 30 m resolution based on SDGSAT-1 TIS data in the Arctic Ocean and detailed comparisons with the S2 data and with the MODIS-derived leads, as well as the cross-comparisons among the three bands. The results are based on a total of 11 TIS data that are grouped into four scenes and have three sub-regions for matching comparison with the S2 (see Figure 1).

**4.1 Comparison of the TIS-detected sea ice lead with Sentinel-2 images**

To examine the reliability of sea ice leads detected in this study, we first carried out comparisons of typical cases under clear sky conditions. The two cases presented below are in the Beaufort Sea near the Canadian Arctic Archipelago. Sea ice leads in this region have typical seasonal variations (Steele et al., 2015). Here, we focused on the leads detected in April 2022 (in the red squares on borders 1 and 2 in Figure 1). Visible images from the S2 MSI green band were collocated with the derived

leads for validation. The three BT maps are displayed in the first row of Figure 8. The detected leads in this study are represented by the white pixels in the following binary maps. For the matched visible images, the albedo of a lead is lower than that of the sea ice surface, so leads are the dark objects in the S2 images. According to the lead study based on S2 data (Muchow et al., 2022), we calculated the normalized brightness and specified that a pixel with a normalized brightness below 0.7 could be a lead, while a pixel with a normalized brightness above 0.07 could be sea ice. Thus, a pixel with a

normalized brightness between 0.07 and 0.7 is considered to have both possibilities. Apparently, our detection results based on the three infrared bands are highly consistent with these visible images. In particular, it is likely that some of the narrow leads detected, with widths of tens of meters, have just begun to form, which are also subtle in 10 m resolution visible images.



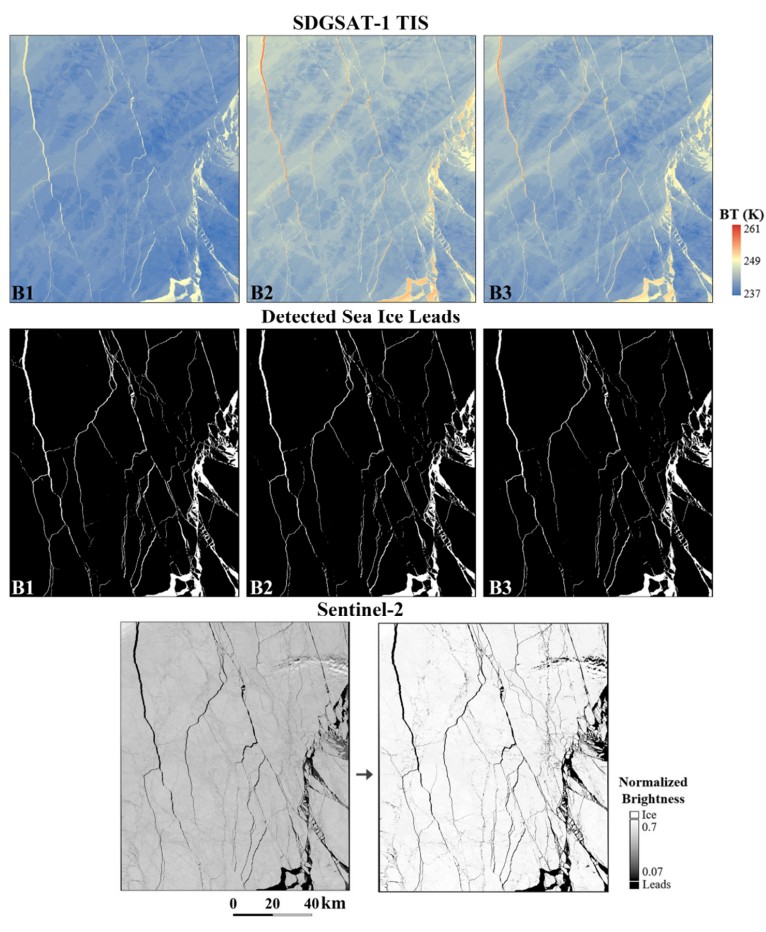

(a)



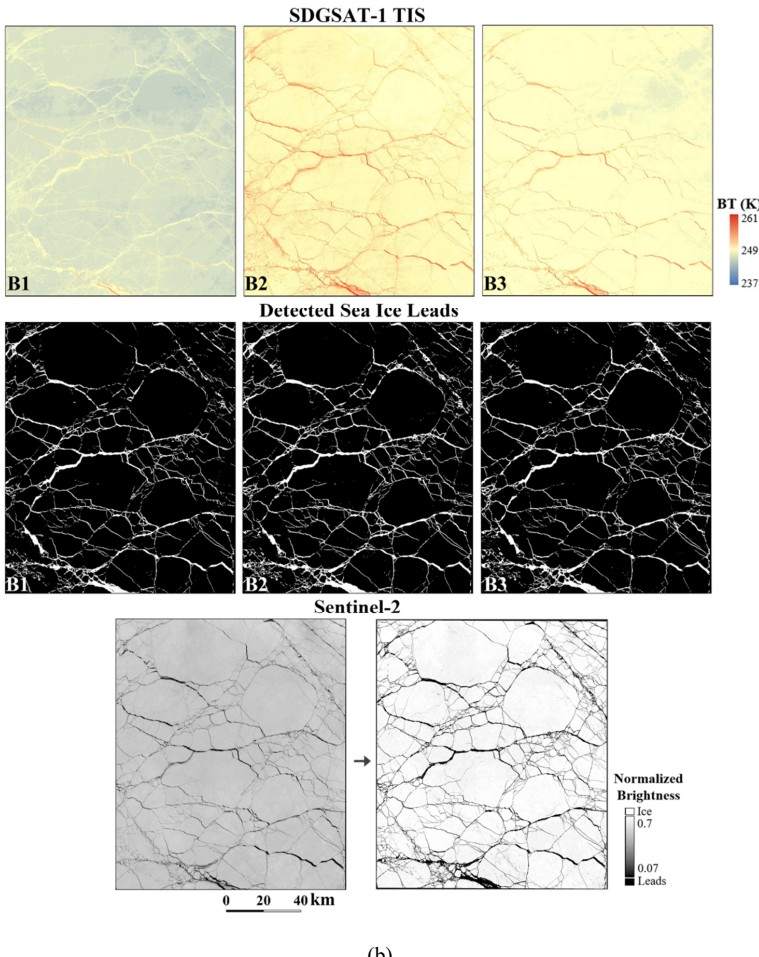

(b)

**Figure 8: Validation of sea ice lead detection based on SDGSAT-1TIS data compared with S2 visible images in the Beaufort Sea, April 2022. The three rows show the BT maps for the B1, B2 and B3 bands, their lead detection results, the S2 images and the normalized brightness (from 0.07 to 0.7), respectively.**
**(a) TIS data acquired at 04:28 UTC and S2 data acquired at 21:00 UTC on April 3, 2022.**
**ID: KX10_TIS_20220403_W128.84_N73.00_202200033226, KX10_TIS_20220403_W132.14_N74.67_202200033227.**
**(b) TIS data acquired at 04:56 UTC and S2 data acquired at 22:42 UTC on April 28, 2022.**
**ID: KX10_TIS_20220428_W147.26_N77.60_202200049406.**



We compared the TIS-based ice leads with the visible images on a pixel-by-pixel basis. The definitions of TP, FP, FN, and TN used are listed in Table 3. Due to the imbalance between the distribution of leads and the ice background, we used three indicators for detection performance: the commission error, the omission error, and the accuracy. The statistics listed in Table 4 for the two cases in the Beaufort Sea show that for all bands, the commission error, omission error, and accuracy are 5.45%, 44.73%, and 96.30%, respectively. For the three bands, the overall accuracy achieves a high level of 96.24%, 96.34% and 96.33%, respectively. The B1 band is satisfactory in terms of an overall commission error of 5.39%, but yields a slightly high miss rate of 46.25%. The omission error mainly attributes to a large FN result, resulting from subtle refrozen leads (covered by thin ice). More specifically, the case on April 3 (shown in Figure 8 (a)) yields a commission error less than 4.60%, while the commission error on April 28 is slightly higher than the former. The reason lies in the differences in the lead distribution and fraction. For the April 28 case (shown in Figure 8 (b)), more leads that undoubtedly exacerbate the difficulty in detection are presented.

Moreover, the BT values recorded by SDGSDAT-1 TIS on these two days were different. Even in the overlapping region of borders 1 and 2 in Figure 1, the BT on April 28 is approximately 5 K higher than that on April 3. This finding may imply a variation in temperature on a short temporal scale in the late spring, allowing the formation of more leads and exhibit more intricate lead nets. On the other hand, a warming environment can reduce the contrast in thermal infrared data, resulting in lower BTA values for leads. The phenomenon is related to different atmospheric conditions, which we further analyze in the Discussion.

By applying this detection method to the TIS data acquired over the Laptev Sea on March 23, 2022 (shown within rectangle 3 in Figure 1), we found a complex situation when compared to the S2 visible image, as shown in Figure 9. The expansive gray feature on the S2 images is more likely to be cloud shadow than leads (McIntire and Simpson, 2002). Detecting leads under this interference is quite difficult since the thermal contrast is far less distinct than that on clean ice surface, as shown in the following BTA maps. Compared to the visible image, the accuracy values for the B1, B2 and B3 bands are 95.53%, 95.43%, and 95.56%, respectively. However, some FP detections remain in the three bands, which are marked by yellow rectangles in the third row. Thus, although this detection based on SDGSAT-1 TIS data show promising applicability, the uncertainty caused by cloud interference remains to be further explored.

**Table 3. Definition of a pixel-by-pixel results comparison for the binary lead detection and the optical images with normalized brightness.**

| | | Normalized brightness of the S2 visible image | |
|---|---|---|---|
| | | **< 0.7** | **> 0.07** |
| **Leads detection** | **1** | TP (True Positive, sea ice leads) | FP (False Positive) |
| | **0** | FN (False Negative) | TN (True Negative, sea ice) |



**Table 4. Lead detection performance based on the TIS data in the Beaufort Sea on April 3 and 28, 2022. Results from each TIS band and from all TIS bands are aggregated**

| | | Commission Error (%) $\dfrac{FP}{TP + FP}$ | Omission Error (%) $\dfrac{FN}{FN + TP}$ | Accuracy (%) $\dfrac{TP + TN}{TP + TN + FP + FN}$ |
|---|---|---|---|---|
| April 3 | B1 | 4.56 | 45.94 | 96.31 |
| | B2 | 3.99 | 47.39 | 96.28 |
| | B3 | 3.93 | 47.73 | 96.26 |
| April 28 | B1 | 6.70 | 46.73 | 96.12 |
| | B2 | 7.28 | 38.86 | 96.44 |
| | B3 | 7.28 | 38.71 | 96.43 |
| **Overall** | B1 | 5.39 | 46.25 | **96.24** |
| | B2 | 5.48 | 43.91 | **96.34** |
| | B3 | 5.47 | 44.04 | **96.33** |
| | **All Bands** | **5.45** | **44.73** | **96.30** |



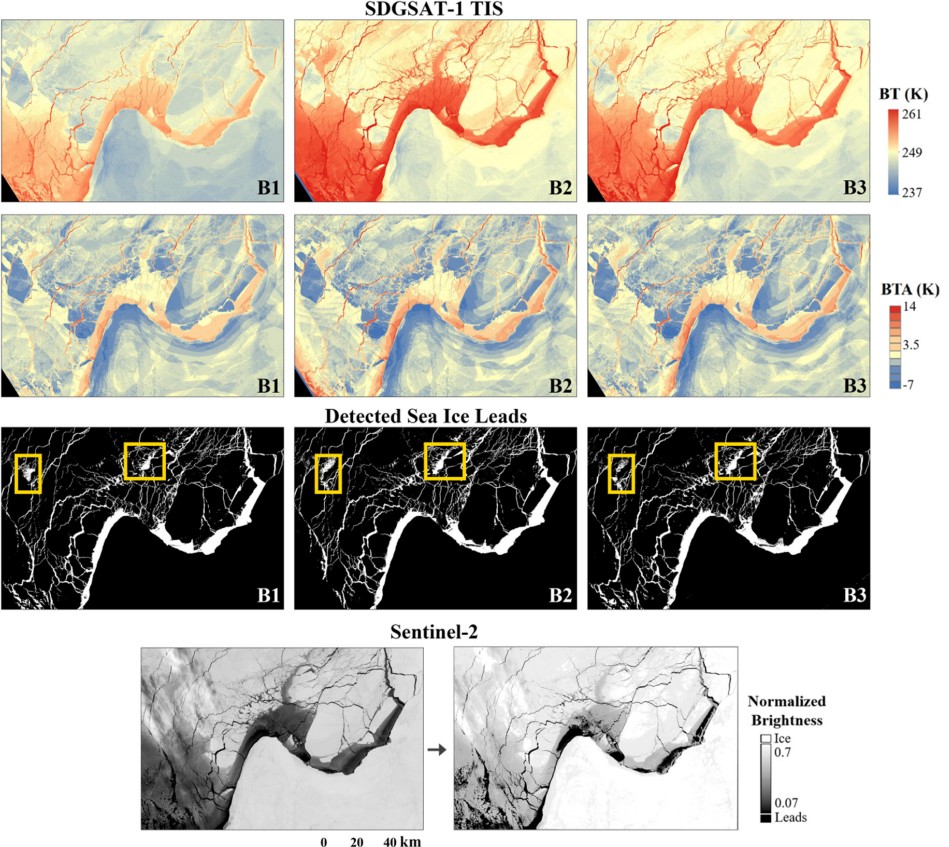

**Figure 9: Application of the lead detection method to the SDGSAT-1 TIS data acquired over the Laptev Sea at 10:53 UTC on March 23, 2022, and comparison with the S2 visible image at 03:55 UTC on the same day (similar illustration to the previous figure). ID: KX10_TIS_20220323_E129.38_N75.60_202200028841 and KX10_TIS_20220323_E133.08_N73.96_202200028843**

**4.2 Cross-comparison of sea ice lead detection based on the three TIS infrared bands**

The three TIS bands all yield good accuracy in lead detection but do present some discrepancies. In this subsection, we performed cross-comparisons of these results to focus on the effectiveness of the three thermal infrared bands in detecting leads. Counting the lead pixels derived from each TIS band, a total of 46,301,986 pixels comprise the consistency detection (co), i.e., a pixel that is detected as ice leads from all three bands. Thus, the additional detection (ad) is calculated (i.e., detected as ice leads by a specific band) using formulas (2) and (3).

$$N_{Bn, ad} = N_{Bn} - N_{co} ,$$ (1)



$P_{Bn,\,ad} = N_{Bn,\,ad} / N_{co} \times 100\%$ ,                                                                     (1)

where N is the total number of pixels; Bn is the infrared band (n = 1, 2, 3); and P is the proportion. The results listed in Table 5 show that the additional detections from the B1, B2 and B3 bands account for 11.46%, 23.30%, and 21.88%, respectively. The fewest leads are detected by the B1 band, while the B2 and B3 bands give similar results.

**Table 5. Statistics of lead pixels detected based on the three infrared bands of the SDGSAT-1 TIS**

| | Leads Pixels Number | Additional detection | |
| --- | --- | --- | --- |
| | | Pixels Number | Proportion |
| B1 | 51,609,678 | 5,307,692 | 11.46% |
| B2 | 57,088,756 | 10,786,770 | 23.30% |
| B3 | 56,430,724 | 10,128,738 | 21.88% |
| **Consistency** | 46,301,986 | | |

To further investigate the discrepancies, we depicted the detections with different colors. As shown in Figure 10, dark red, orange and dark blue colors mark the leads detected by the B1, B2, and B3 bands, respectively. The discrepancies primarily occur in the margins of leads. In Figure 10 (a), the comparisons in the second (B1 vs. B2) and fourth columns (B1 vs. B3)

indicate that the B1-derived leads are generally less than the B2 and B3 bands. The third column (B2 vs. B3) presents only a small number of spatial variations, probably due to local temperature gradients. Thus, it can be concluded that the TIS B2 and B3 bands yield almost comparable performances in detecting sea ice leads. These two infrared radiance bands, applied as the two split windows for temperature retrieval, are widely used in infrared sensors, e.g., the currently in-orbit Gaofen-5 (GF-5) Visual and Infrared Multispectral Sensor (VIMS), Landsat-8 TIRS, Landsat-9 TIRS-2, and Terra/Aqua MODIS.

However, the example in Figure 10 (b) shows a different scenario. There are more dark red pixels in the cross-comparisons. In particular, some dark red pixels (marked by the black squares) are only presented in the B1 band results, while the B2 and B3 bands almost lose all this information. Figure 10 (c) shows the S2 visible images acquired in the same location, where the lead characteristics are evident (marked by white squares). Indeed, the BT and BTA maps found no apparent differences in the lead thermal characteristics. It is speculated that the missing data in the B2 and B3 bands may result from interference

induced by strip noise, which is particularly pronounced in the two bands (a similar phenomenon is also presented in the split-window channels of MODIS and Landsat 8 TIS). Regardless, this example suggests that using the TIS B1 band appears to achieve unexpected effects in the presence of interference in B2 and B3 data. In other words, the B1 band can be complementary to the two split-window bands. Thus, combining the results of the three bands is beneficial for resolving the leads with better accuracy.



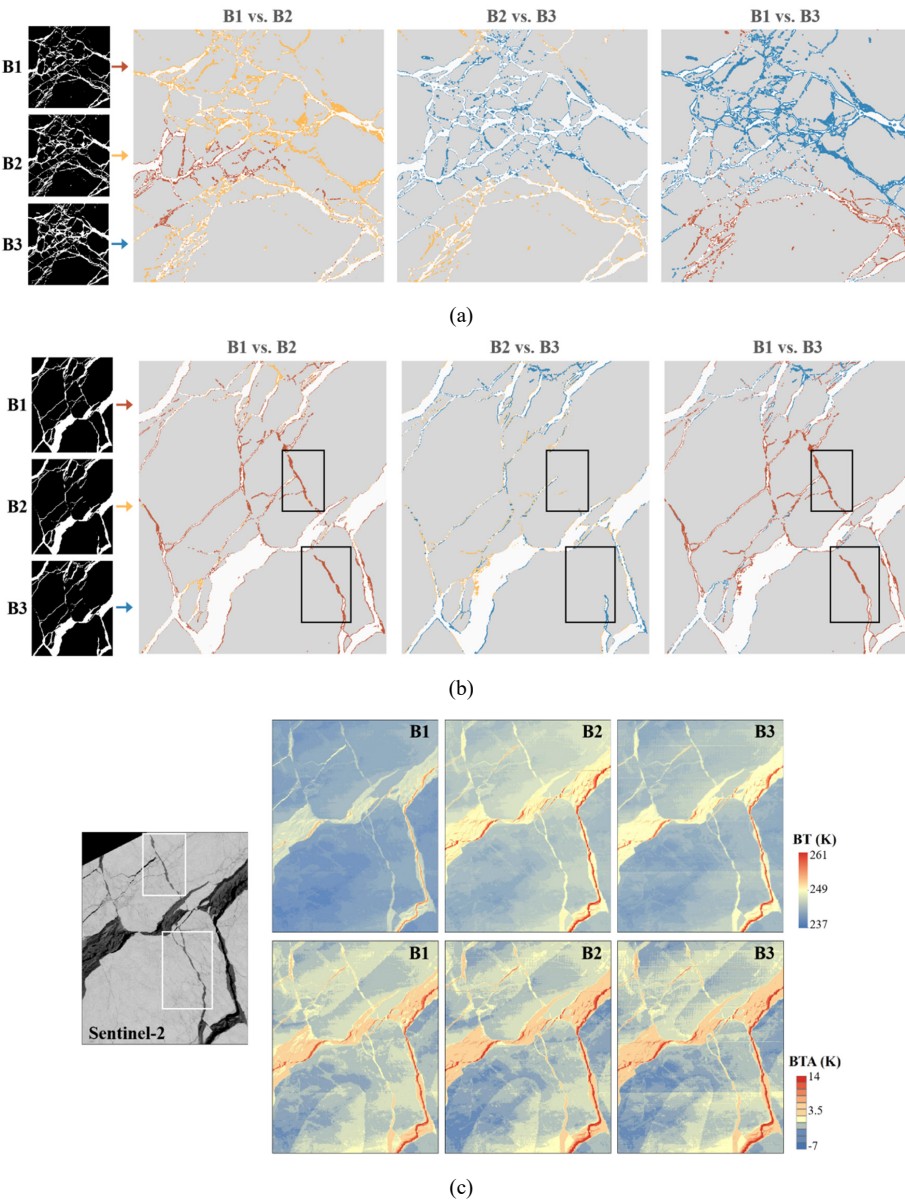

Figure 10: Cross-comparisons between the detections among the three TIS infrared bands. The first column in (a) and (b) shows the lead detection by the three bands. The following three columns are the pairwise comparisons, where the dark red, orange, and



dark blue pixels represent the B1-derived and B2-derived B3-derived results, respectively. White is the consistency detection, and the light gray background is the ice surface. Acquired from the same location as (b), the left panel in (c) shows the S2 image as a reference, and the two parallel rows of panels are the BT and BTA maps for the three bands. (a) TIS data acquired at 04:56 UTC
on April 28. ID: KX10_TIS_20220428_W147.26_N77.60_202200049406. (b) and (c) TIS data acquired at 04:28 UTC on April 3. ID: KX10_TIS_20220403_W132.14_N74.67_202200033227. (c) S2 data acquired at 21:00 UTC on April 3.

### 4.3 Comparison of the TIS-derived sea ice leads with the MODIS

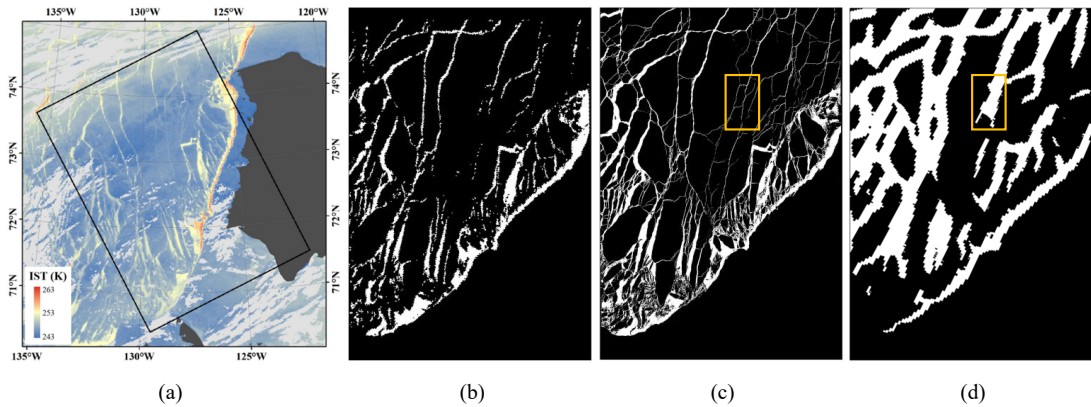

(a)                           (b)                           (c)                           (d)

**Figure 11: Comparisons between lead detection based on MODIS data and SDGSAT-1 TIS data in the Beaufort Sea on April 3, 2022. (a) MODIS IST products, where the clouds are off-white, the land is dark gray, and the overlaid black border denotes a
coverage for (b), (c) and (d). (b) Leads at 1 km resolution derived by MODIS IST product. (c) Leads at 30 m resolution derived from the combined result of SDGSAT-1 TIS B1, B2 and B3 bands. ID: KX10_TIS_20220403_W126.10_N71.30_202200033225, KX10_TIS_20220403_W128.84_N73.00_202200033226, KX10_TIS_20220403_W132.14_N74.67_202200033227. (d) Leads at 1 km resolution derived by Hoffman et al. (2022b).**

**Table 6. Statistics of the lead area estimated from the MODIS IST data and the SDGSAT-1 TIS data.**

| | | Sea ice lead area (km$^2$) | | | Additional lead area by the TIS than by Hoffman et al. (2022b) (km$^2$) |
|---|---|---|---|---|---|
| | | MODIS IST | SDGSAT-1 TIS | TIS/MODIS IST | |
| 1 | Beaufort Sea on April 3 | 13,302 | 15,362 | 1.15 | 5,679 |
| 2 | Beaufort Sea on April 28 | 1,620 | 10,500 | 6.48 | 4,590 |
| 3 | Laptev Sea on March 23 | 4,103 | 4,519 | 1.10 | 1,462 |
| 4 | Laptev Sea on March 23 | 3,887 | 3,936 | 1.01 | 2,415 |



| **Total** | **22,911** | **34,318** | **1.50** | **14,145** |
|---|---|---|---|---|

We further compared the TIS-derived ice leads with the MODIS data at a moderate resolution. The previously developed method (Qu et al., 2021) was applied to detect the leads based on the MODIS IST data. The IST products in March and April 2022 were first mosaicked (as one case in the Beaufort Sea on April 3, 2022 presented in Figure 11 (a)) and then applied to the binary segmentation by a BTA threshold of 1.2 K to derive the lead data (the corresponding result is shown in Figure 11

(b)). Simultaneously, we combined the lead detections based on the three TIS bands, and the result is one binary map containing the most leads (see Figure 11 (c)). There is a significant difference between the high- and moderate-resolution results. The TIS resolves more lead details, e.g., the narrow leads connecting those wide ones. Notably, some of the leads in Figure 11 (c) are even obtained in places that are considered clouds by the MODIS cloud mask. For the sake of unity, no comparisons were made in the cloud-masked pixels.

Correspondingly, the lead area was calculated by the two datasets in the same regions. For the four regions shown in Figure 1, the comparisons of the ice lead areas are listed in Table 6. The area estimated from the TIS data is significantly larger than that estimated from MODIS IST, with the total area exceeding the latter by more than half. In particular, the difference of the lead area between the TIS and MODIS reaches its maximum in the comparison of the case in the Beaufort Sea on April 28. The leads detected by the TIS are 10,500 $km^2$ in area (with the B1, B2 and B3 bands of 7,752 $km^2$, 9,346 km2 and 8,973 $km^2$,

respectively). This could be attributed to the temperature variations on a short temporal scale. The IST increases to approximately 260 K in the Beaufort Sea on April 28, far beyond the general IST from 240 K to 250 K for the study area (also see Figure 12 (a)). Consequently, the reduced thermal contrast of leads severely limits the ability of MODIS to detect leads. In contrast, the high-resolution imaging capability and high sensitivity of the SDGSAT-1 TIS can present more significant thermal contrasts of leads and ice.

Furthermore, Hoffman et al. (2022b) published the lead dataset since 2002 for the season between November through April, which were detected by the U-net model (Hoffman et al., 2021) based on MODIS 11 μm thermal imagery. The spatial resolution of the dataset is 1 km, which were reported as daily aggregated detection frequency. As showed in Figure 11 (d), lead widths and areas are significantly larger, especially as small leads in close are identified as an entire large lead (see the orange squares marked in Figure 11 (c) and (d)). Given that this dataset is not appropriate for direct estimation of lead area,

we used it as a mask and only calculated the area for the TIS-derived leads beyond this mask (i.e., the additional area). The statistics are presented in the last column of Table 6. A total of additional leads derived by the TIS is 14,145 $km^2$ compared to that derived by Hoffman et al. (2022b), which is generally in line with the result of the TIS and MODIS IST comparison (11,407 $km^2$). This result suggests that the overlooked narrow leads by the moderate-resolution sensor are predominant. Since the turbulent exchange efficiency over the leads is very strongly determined by their width and area, the lead

observation at a high spatial resolution is essential to achieve an accurate lead width parameterization and to further estimate their thermal effects. These comparisons with moderate-resolution sensor prove that the TIS is a competitive sensor for detecting sea ice leads in polar regions.



## 5 Discussion

Based on the TIS onboard SDGSAT-1 with the high-resolution thermal infrared data available, we successfully detected sea
ice leads in the Arctic for the first time at 30 m resolution, achieving good results in terms of the detection accuracy,
adaptability and ability to characterize narrow details. In this section, we focus on discussing the influence of different
atmospheric conditions on uncertainties in TIS leads observation and analyzing the leads properties revealed in the detection.

### 5.1 Atmospheric influences on leads detection by TIS three bands

First, as an important constraint on the Arctic lead detection, it is necessary to consider the impact of cloud interference.
Although cloudy conditions are prevalent in the Arctic (see the large white area in the MODIS daily IST product shown in
Figure 12 (a)), due to the unavailable cloud products synchronised with the SDGSAT-1 TIS, this study only demonstrates the
lead detection under clear sky conditions. However, we agree with the view of Hoffman et al. (2019) that using cloud mask
products in ice leads detection would produce omissions as a result of incomplete cloud information. They reclassed the
MODIS cloud mask products to eliminate omission errors and assumed that a lead pixel would have a BT less than 271 K
(otherwise, it would be cloud). We manually collected cloud-less data and used the BT filter, which rejected the pixels with
BT values greater than approximately 245 K. Therefore, clouds likely had a relatively small impact on the results of this
study, but the impact of ice clouds still remains. In the future, with the availability of the SDGSAT-1 cloud product, we can
further investigate the lead detection for the cloudy conditions.

Apart from cloud interference, other atmospheric components also affect lead detection. The TIS B2 and B3 bands, the two
atmospheric windows, are nearly transparent to the atmosphere, and therefore, to some extent, they can obtain surface
radiance independent of the atmosphere. In contrast, the B1 band, as an absorption channel for ozone (Wan and Li, 1997),
has attenuation in the atmosphere. As a result, the B1 band data present different temperature gradients from the other two
bands, particularly pronounced at high latitudes (Prabhakara and Dalu, 1976). In addition, short-term temperature variations
also affect the temperature contrast for the three thermal infrared bands and thus the detection of leads. Since the at-sensor
BT data used in this study were not corrected for atmospheric radiation, this temperature variation results from a
combination of sea ice radiation and atmospheric radiation. As displayed in Figure 12, the temperature of sea ice surface and
2 m air, and the ozone resolution present significant temporal and regional variations. Both the air and ice temperatures
gradually increase and show similar spatial patterns. The ozone resolution is high in the Laptev Sea and the Beaufort Sea,
and its distribution changes rapidly on a monthly scale. We analyzed the sensitivity of lead temperature characteristics to
these factors. First, based on the detected leads, we extracted the BT and BTA data only for those lead pixels and allocated
them to the geographic grids at 30 km. Then, regression analysis was conducted to find the relation between the thermal
characteristics and IST, air temperature, or ozone resolution, as listed in the Table 7.

In general, the BT data from the TIS three bands have significant positive correlations with the IST data and air temperature.
Although the upward slope of the BT data with ice and air temperatures for the B1 band is smaller than that for B2 and B3



bands, the high correlation (of 0.72 with the IST and 0.68 with the air temperature) demonstrates its effectiveness as a thermal infrared band for lead detection. On the other hand, changes in IST have only small negative correlation with the BTA values of leads. While changes in air temperature are more likely to diminish the thermal contrast of the leads, which have less effect on the B1 band and more effect on the B2 band. These results imply that atmospheric correction and ice temperature retrieval of TIS thermal data could be effective approaches to improve the robustness of lead detection. For the

three thermal infrared bands of the TIS, the B1 band may not be as sensitive to temperature variations as the B2 and B3 bands.

In our expectations, only the BT data from the B1 band have a negative correlation with ozone, with the correlation of -0.62. Evidently, only the B1 band radiance is strongly absorbed by ozone, which also explains why the B1 band gives the lowest BT values for the presented cases in this paper. With respect to the BTA values of the leads, none of them shows significant

correlations with the ozone resolution. This finding implies that although ozone affects the B1 band temperature measurement, it barely weakens the thermal contrast for lead detection.

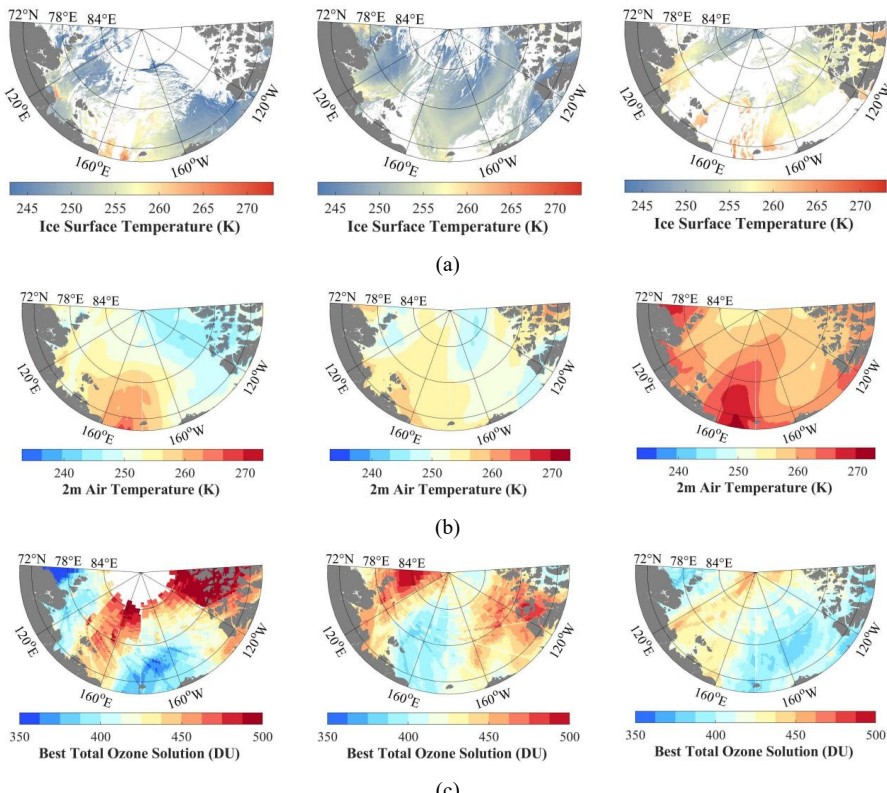





**Figure 12: (a) MODIS IST data, (b) ERA5 2 m air temperature data, and (c) OMTO3 ozone resolution data, in which the three columns are on March 23, April 4 and 28 2022 respectively.**

**Table 7. Correlation between the thermal characteristics of SDGSAT-1 TIS three infrared bands and the IST, 2 m air temperature, and ozone resolution, with the slope of the regression fitting in brackets.**

|  | B1 band | | B2 band | | B3 band | |
|---|---|---|---|---|---|---|
|  | BT value | BTA value | BT value | BTA value | BT value | BTA value |
| **IST** | **0.72** (0.54) | -0.18 (-0.11) | **0.64** (0.70) | -0.36 (-0.26) | **0.71** (0.64) | -0.29 (-0.20) |
| **Air temperature** | **0.68** (0.53) | -0.39 (-0.26) | **0.68** (0.75) | -0.55 (-0.42) | **0.68** (0.63) | -0.50 (-0.25) |
| **Ozone resolution** | **-0.62** (-0.07) | 0.14 (-0.01) | -0.49 (-0.08) | 0.19 (0.20) | -0.59 (-0.08) | 0.24 (0.02) |

**5.2 Sea ice lead property resolved by TIS**

Due to thermal infrared imaging having long been limited to the moderate resolution of kilometers, it is difficult to confirm either the widths of narrow leads or the variations inside them. The detection at 30 m resolution presents thermal variations

inside leads, as an interesting case shown in Figure 13, which was acquired on April 3 over the Beaufort Sea. The TIS data present a noticeable transition zone (with a BTA value less than 2 K), which is likely seawater intrusion into the sea ice, while the lead in the center (with a BTA value greater than 3 K) was just opening. As the method of this study aims to extract all potential leads, the entire transition zone was marked as an ice lead. This is reasonable, as a previous study (Qu et al., 2020) used 1.2 K for potential leads, 1.5 K for general leads and 2 K for open water discrimination. Given that the binary

segmentation in this paper applies a 1.8 K threshold, it again indicates that the thermal information obtained by SDGSAT-1 TIS presents a greater thermal contrast.

Broadly speaking, fracture zones covered by thin ice and intruded by seawater are also considered leads. For other supporting evidence, we incorporated the S1A SAR images acquired on the same day. The dual-polarization data were radiometrically calibrated, and a false-color composition was performed by assigning the HH, the subtraction of HH by HV

and the HV images to the red, green and blue channels, respectively. The HH and HV SAR data and the false-color composite images are shown in the panel at the bottom right of Figure 13. The overall backscatter values for the HH and HV data are low. However, in the transition zone of the lead, the backscatter values of the HH and HV data differ considerably, while the opening in the center has low backscatter values. Accordingly, the transition zone presents a yellowish color in the false-color composite image, while the opening lead is darker. Therefore, the leads detected in this paper based on thermal

contrast agree with the properties resolved by the polarization differences in SAR. Regarding the application of SAR data to lead detection, its applicability to local sea ice conditions remains to be further explored.

In addition, contours of multiyear ice with high backscatter values that are observed in SAR images are similar to some negative BTA features. In particular, the B2 band is more sensitive to such surface information because various types of sea ice have different emissivity and produce different BT values. Thanks to the high-resolution characterization of the



SDGSAT-1 TIS and the accurate radiometric measurement, it is possible to reveal the sea ice properties (inside the leads and on the ice surface). For sea ice with a high-temperature characteristic (possibly thicker or resulting from local temperature gradients), its BTA can be too similar to a lead to be distinguished (Key et al., 1993), which is why we preferred a BT filter after the lead segmentation in this study.

The special case shown in Figure 10 arouses our interest in why the B2 (b) and B3 (c) bands missed some leads. As

described in Section 4.2, the strip noise also affects lead detection. The strip noise is the most severe in the B3 band and secondary in the B2 band, while it is absent in the B1 band. This difference occurs because when the TIS overpasses a homogeneous surface, which covers sea ice with a low radiance signal in this case, each detector gives a different noise bias (Corsini et al., 2000). This phenomenon is even more severe for detectors with higher signal-to-noise ratios. Likewise, to overcome the strip noise, it is necessary to apply the BT filter and use an appropriate threshold. The thresholds determined

by iterative selection were 243.93 K, 248.02 K and 247.14 K for B1, B2 and B3 in this case, respectively. Consequently, high thresholds of B2 and B3 resulted in omissions of some lead details in the detection. From this perspective, residual noise in high-resolution thermal infrared images may have an impact on the lead detection based on the TIS B2 and B3 bands; while the B1 band is less affected due to its relatively low sensitivity. On the other hand, as the TIS data available within the scope of this paper is relatively limited, these individual case studies presented may be weak in terms of

generalizability. In the future, with support by a large amount of data, we will work on a method that can overcome a variety of interferences for application to SDGSAT-1 TIS data to more accurately detect sea ice leads.

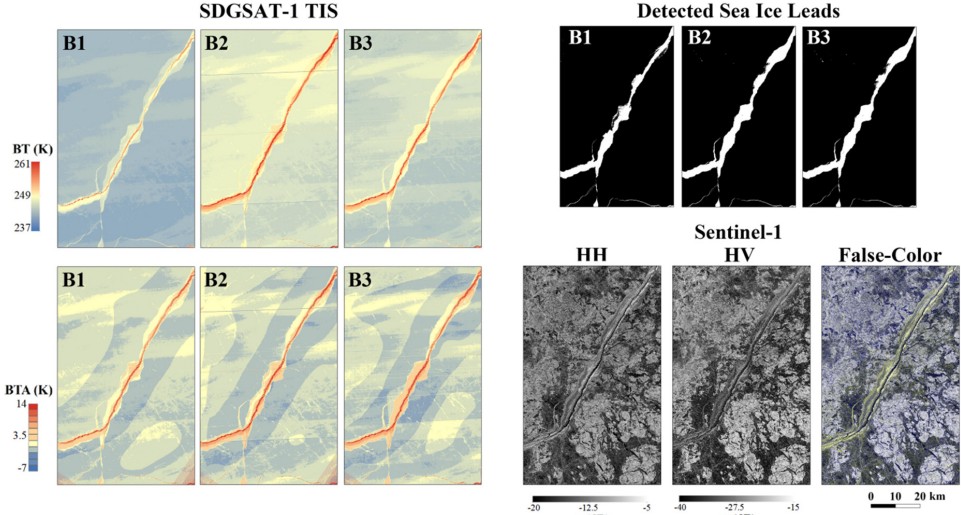

**Figure 13: Comprehensive analysis for lead properties in the Beaufort Sea based on SDGSAT-1 TIS data acquired at 04:28 UTC on April 28 and the derived leads, along with the matched S1A data at 15:52 UTC on the same day. Two parallel rows of panels on**
**the left show the BT and BTA maps for the three bands of SDGSAT-1 TIS. The first row on the right panel shows the leads**



**detected in this study. The panel at the bottom right displays the matched S1A HH, HV and false-color composite images that present recognizable leads. SDGSAT-1 TIS data ID: KX10_TIS_20220403_W132.14_N74.67_202200033227.**

**6 Summary and conclusion**

Over the past decades, the increasing Arctic temperatures and rapid retreat of sea ice include have profound implications for
both the Arctic and the extra-polar climate and ecosystems. Sea ice leads are a key factor influencing heat exchange between the ocean and the overlying atmosphere. However, lead observations based on thermal infrared remote sensing have long been limited to moderate resolutions on a kilometer scale, making it challenging to resolve lead details and resulting in inadequate estimates for ice lead parameters. There is an urgent need to develop fine-scale datasets of leads.

The recently launched SDGSAT-1 provides an emerging opportunity to detect leads at high spatial resolutions up to 30 m by
its onboard payload the TIS. This paper demonstrates the feasibility of using the three TIS infrared bands for detecting ice leads in the Arctic Ocean. We proposed a method that combines binary segmentation with the BT filter to detect leads by TIS data in three bands. The detection results show great details of the narrow leads of tens of meters in width, as well as high accuracy. For example, in the Beaufort Sea case in April 2022, the overall accuracies are 96.24%, 96.34% and 96.33% for the B1, B2 and B3 bands, respectively, compared with the S2 visible images at 10 m resolution. Because more narrow
leads are detected by the TIS, the TIS-derived lead areas are more than half of the results based on the MODIS IST data at a 1 km resolution in the 11 collected cases. This finding indicates that more leads exist in the Arctic Ocean than we have ever observed. These narrow leads beyond our expectations would allow for more heat exchange (Marcq and Weiss, 2012). Therefore, the TIS sensor is expected to improve the lead representation, which is crucial for climate models.

The cross-comparisons among the TIS three infrared bands suggest that the B2 and B3 bands have similar performances in
detecting leads, whereas the B2 band yields the best performance among the three bands. Although the B1 band is less commonly used in thermal infrared measurements, the leads detected by the B1 band can be complementary to the other two on some occasions. We suggest using the combined results of the leads detected from the three TIS bands.

Furthermore, the analysis of the correlation between the detected leads and temperature suggests that B1 (both its BT and BTA data) is not sensitive to the variations in surface and near-surface temperature. Although ozone in the atmosphere
absorbs B1 band radiance, ozone has little impact on the detection of ice leads by the B1 band. The different sensitivity of the B1 band to surface information and atmospheric conditions from the other two bands produces an unexpected performance in sea ice lead detection. Regarding the variations inside the leads, an analysis incorporating the S1A data agrees with the lead properties revealed by our results, but the threshold currently used does classify the transition zone as an ice lead. Thanks to the sufficiently high resolution of the SDGSAT-1 TIS, it is expected to provide crucial data for the
analysis of lead formation and refreezing based on sequential thermal infrared data, an aspect that deserves future attention.

This study is the first to investigate the detection of sea ice leads by spaceborne thermal infrared remote sensing at a high spatial resolution of tens of meters. The results demonstrate that the TIS onboard SDGAST-1 has excellent potential for detecting sea ice leads (as well as possible IST) in polar regions. Along with more TIS data acquired in the Arctic throughout



an entire year and the development of near-real-time cloud product, we can expect to investigate its capability for the
detection of ice leads in different seasons. By combining this data with more diverse datasets of sea ice, we wish to provide
insights into the contribution of leads to Arctic sea ice dynamics in an effort to support SDG 13: climate action.

Furthermore, our investigation suggests that the TIS has high sensitivity to surface temperature changes, yielding great
temperature contrasts to distinguish anomalies. Encouragingly, the sensor should also have great potentials on supporting
research related to other SDG indicators, for instance, SDG 7 in investigating the urban heat island effect to promote green
cities, SDG 14 in monitoring wastewater discharge in coastal zone to protect marine ecosystem, and SDG 13 for monitoring
wildfire and heatwave events to understand impacts of climate change. The TIS, together with other two sensors onboard
SDGSAT-1, are expected to provide more valuable data to facilitate a global approach to the SDGs.

**Data availability**

The SDGSAT-1 TIS data can be acquired from the International Research Center of Big Data for Substantial Development
Goals (www.sdgsat.ac.cn/, last access: 20 December 2022). The S2 data are available on the United States Geological
Survey website (https://www.usgs.gov/, last access: 20 December 2022). The S1 data are accessed from the Copernicus
Open Access Hub (https://scihub.copernicus.eu/dhus/#/home/, last access: 20 December 2022). The MOD29P1D product
can be acquired from the National Snow and Ice Data Center (https://nsidc.org/, last access: 20 December 2022; Hall and
Riggs, 2021). The ERA5 datasets are available on European Centre for Medium-Range Weather Forecasts
(https://cds.climate.copernicus.eu/, last access: 20 December 2022; Hersbach et al., 2018). The OMTO3 products can be
acquired from the NASA Earth Observation Data web (https://disc.gsfc.nasa.gov/, last access: 20 December 2022; Bhartia,
2012). The sea ice lead dataset published by Hoffman et al. (2022b) are available from
https://doi.org/10.5061/dryad.79cnp5hz2 (last access: 20 December 2022).

**Author contribution**

XML conceived the idea and designed the research. YQ developed the method and conducted the experiments. HDG
provided insightful suggestions and discussions. All authors contributed to writing the manuscript.

**Competing interests**

The authors declare that no conflicts of interest or personal relationships influenced the work reported in this paper.



**Acknowledgements**

It is acknowledged that the SDGSAT-1 data are provided by the International Research Center of Big Data for Substantial Development Goals, and other data used is also acknowledged. The authors particularly thank the team led by Dr. Hongyu Chen and Bihong Fu in Innovation Academy of Microsatellites of CAS and the team led by Dr. Fansheng Chen in Shanghai Institute of Technical Physics of CAS for their great efforts on development of the SDGSAT-1 satellite and the onboard TIS payload. Mr. Weixing Wang from the SDGSAT-1 ground segment team of Aerospace Information Research Institute of

CAS provides great support on the TIS data acquisitions over the Arctic. Dr. Yonghong Hu from the SDGSAT-1 calibration team of AIR, CAS explained the calibration of the TIS data.

**Funding**

The study is partially supported by the National Science Fund for Distinguished Young Scholars (No. 42025605).

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
