# Peer review of "Spaceborne thermal infrared observations of Arctic sea ice leads at 30 m resolution"

_EGUsphere, 2022_

## Referee Comment (RC2)

**Spaceborne thermal infrared observations of Arctic sea ice leads at 30 m resolution**

Yujia Qiu, Xiao-Ming Li, Huadong Guo

**13 Feb 2023**

**General comments**

The authors have developed a method to map Arctic sea ice leads using Thermal Infrared Spectrometer (TIS) data onboard recently (Nov 2021) launched Chinese Sustainable Development Science Satellite 1 (SDGSAT-1). TIS has three IR bands: B1 (8.0-10.5 μm), B2 (10.3-11.3 μm) and B3 (11.5-12.5 μm), with 30 m resolution and 300 km swath width. The TIS instrument provides TIR data at much finer resolution than MODIS (1 km) and VIIRS (750 m), and somewhat finer than Landsat 8/9 (100 m). Optical and NIR data at comparable or even finer resolution are available from many sensors (e.g. Sentinel-2, Landsat 8/9), but only TIR data enables to retrieve sea ice properties regardless of daylight conditions.

The method for the Arctic sea ice leads mapping is developed using 11 TIS images acquired over the Laptev and Beaufort Seas in Match and April 2022. The general lead detection method was same for all three TIS bands with some variable parametrization, i.e. all three bands were not used together for the lead detection, but in some cases the resulting three lead maps were combined (if I understood correctly, please see my comment later). The same TIS images were also processed to the leads maps (binary map: lead (open water or thin ice) or sea ice. The leads by the three TIS bands were compared to each other, and to lead detection maps by Sentinel-2 band 2 (green band) data and MODIS daily IST data. The Sentinel-2 lead detection was based on study by Muchow et al. (20121) and the MODIS lead detection to (Qu et al. 2021). A case study was also conducted were a TIS lead map was compared to Sentinel-1 dual-polarization SAR image. Possible atmospheric influences to the TIS lead detection were investigated using ERA5 air temperature and OMI ozone total column data. The TIS band 1 covers the ozone absorption band.

In general, the TIS lead detection seemed work fine, and against the S-2 lead maps there was 96.3% pixel based match (authors used TIS accuracy here). Compared with the MODIS lead map, the TIS map presents more leads with width less than hundreds of meters. All three TIS bands showed similar performances in detecting leads. The B1 band can be complementary to the other two bands, as the temperature sensitivity is different from the other two, benefiting better detection by combining the three bands. Arctic lead maps with fine resolution (e.g. 30 m) allows to observe narrow leads which are undetected in the MODIS/VIIRS products, and estimate their contribution to the overall Arctic lead fraction.

I think the study set up with data acquisitions and data processing is rather well conducted, and the TIS lead detection method could be generally applicable for a large number of TIS images, but I think it is not sure as it is based on small amount of data. Further, I don't think it is meaningful to develop lead detection methods for the three TIS bands separately, and compare the results. You should develop the best possible lead detection algorithm for the TIS data (having as input all bands or just two/one), and only present this in the paper. In the following I have also some other major comments to the papers and suggestions for possible improvements. These are followed by miscellaneous specific ones.

The review and discussion on related previous studies is well conducted, e.g. different methods for lead detection are nicely discussed. However, your review could include also following new study:

Q. Wang, M. Shokr, S. Chen, Z. Zheng, X. Cheng and F. Hui, "Winter Sea-Ice Lead Detection in Arctic Using FY-3D MERSI-II Data," in *IEEE Geoscience and Remote Sensing Letters*, vol. 19, pp. 1-5, 2022, Art no. 7005105, doi: 10.1109/LGRS.2022.3223689.

lead map with 250 m resolution

There are many studies with lead detection using SAR and altimeter data, and it is fine to give only few examples as you have done. Related to (Murashkin and Spreen, 2019) include also reference:

Dmitrii Murashkin, Gunnar Spreen, Marcus Huntemann, and Wolfgang Dierking, "Method for detection of leads from Sentinel-1 SAR images," Annals of Glaciology, vol. 59, no. 76, pp. 124–136, 2018

Could you use their method for automatic lead detection in your S-1 SAR imagery? This would allow better utilization of the SAR imagery as comparison data.

Your review includes only one study where Landsat data are used for the lead detection (Qu et al. 2019). Are there any other Landsat studies? Please check. One relevant study here could be:

Cáceres, A.; Schwarz, E.; Aldenhoff,W. Landsat-8 Sea Ice Classification Using Deep Neural Networks. Remote Sens. 2022, 14, 1975. https://doi.org/10.3390/rs14091975

Sentinel-2 lead study by (Muchow et al. 2022) must be discussed in Introduction, it is now mentioned later in Section 4.1.

You could also summarize currently publicly available Arctic lead products with their time spans, seasonal coverages (full year or only winter season) and spatial resolutions.

Section 2 Data could be changed to "Data and pre-processing", i.e. to include all data processings before analyses and lead detection method development, e.g. move Section 3.1 (Pre-processing of TIS data) to Section 2.

Detailed descriptions of TIS instrument and its data should moved from Introduction to Section 2.1. In the following are some questions on the TIS data:

What is the size of the TIS image along track?

What is the main intended application of the band 1?

What is the resolution of the TIS data in K?

Is there yet IST product from bands 2 and 3? Under development?

Are there any TIS cloud masking algorithms or products?

BTA threshold for the lead detection was manually selected (here 1.8 K). Are you sure this is really applicable for a large amount of TIS images acquired in various sea ice and atmospheric conditions? Why did you not develop an automatic selection method for the BTA threshold, as you did for the BT threshold? I think is rather serious potential flaw in your lead detection method. You should really have an automatic BTA threshold selection method.

In general, we could have better reliability on your TIS lead detection method if more TIS data had been used in its development. Can you add more TIS images to your study?

The TIS lead map is evaluated against the Sentinel-2 lead map, and their agreement is very high, 96.3%. You talk about accuracy of the TIS lead map based on this comparison, but this comparison really don't give an absolute accuracy of the TIS map, as I don't think your Sentinel-2 lead map is error free. For determination of accuracy in-situ or airborne data are needed.

TIS strip noise is discussed first time under Results. It should be mentioned when TIS sensor and data properties are described under Section 2.

It would be very interesting see lead map comparison between TIS (30 m) and Landsat (100 m), can you add this to your paper? We could see how much 30 vs. 100 m resolution matters in the lead detection.

Finally, under 'Summary and conclusion" Section you discuss more about future research goals, and will be there an operational TIS lead product?

**Specific comments**

Abstract

"unresolvable ice leads"; change to "unresolvable sea ice leads" or to "unresolvable leads"

1. Introduction

line 28: "under wind and water stresses"

  better "under wind and ocean stresses"

l. 33: "During winter, newly opened leads are the main source of ice production, brine rejection, and turbulent heat loss to the atmosphere"

  Is it leads or polynyas in the whole Arctic scale? Please check. Anyway polynyas could be mentioned in this context.

l. 47: "Other studies also applied active and passive microwave data to lead detection with the advantage that microwave wavelengths are transparent to cloud cover; however, either the data resolution is too coarse"

  Too coarse to what? Detect smaller leads? In some application of the lead data?

l. 57: "Essentially, IST data, which are generally retrieved by the split-window technique (Key et al., 1997), are less accurate under the presence of melt ponds and leads because the lower emissivity (0.96 compared to 0.99) can cause a difference in the retrieved temperature"

  Lower emissivity of what? Water?

l. 61: "They detected leads for January through April over the period of 2003 to 2018, presenting a lower estimation for the lead area compared with the results in Willmes and Heinemann (2015c); the reason is the difference in spatial resolutions of the lead datasets."

  Give resolutions of these datasets in the text.

l. 67: "Qu et al. (2021) proposed a modified algorithm

  Modified from what? From Hoffman et al?

l. 94: "To date, the TIS has acquired substantial high-resolution thermal infrared data from the critical seas in the Arctic"

  How do you define what is a critical area in the Arctic? Explain in the text.

l. 102: "This study is the first to observe Arctic sea ice leads at 30 m resolution"

Table 1 must include references to the data it present.

Figure 1 should include acquisition times for the rectangles 1-4. Describe that visible images come from Sentinel-2.

2. Data

l. 130: "Considering the benefit of incorporating three thermal infrared bands for observation, Thus, the three bands of SDGSAT-1 TIS data are used"

  You can remove 'thus' from the sentence.

l. 131: "The georeferenced level-4 TIS data"

  What do you mean by level-4 data here?

Give some refences to Sentinel-1 and -2 sensors and data.

l. 139: "given that the visible spectrum centered at 560 nm gives a good effect (König et al., 2019) for a scene containing sea ice and seawater."

What is this 'effect'? Good discrimination between sea ice and water?

l. 147: "we collected S1A dual-polarization data in the Beaufort Sea on April 3 and 28, 2022 (see Table 2).

Table 2 shows S-1 data only on 3 Apr.

Table 2: S-1 resolution is not 40 m, it is around 100 m, the pixel size is 40 m. Explain what are h07/08 etc. under MODIS column.

l. 163: "air temperature) data available by every 6 hours"

Your ERA5 reference shows that data is hourly.

OMI ozone: give some reference also for the ozone retrieval, in addition to the product reference.

**3. Method**

l. 195: "In addition to the leads presenting as distinct yellow and red colors on the BT map,

Give also temperature ranges these colors represent.

Give some references to the BTA based lead detection in Section 3.2, as this method has been used in many studies using MODIS and VIIRS data.

l. 208: "By collecting eight TIS data acquired between April 3 and April 28"

There is seven TIS images in Table 2.

l. 222: "Previous methods applied a variety of BTA thresholds"

should be "Previous studies", and give also references.

Figure 6 and 7: give TIS band used in the figures.

l. 268: "Finally, the binary detection of leads at a 30 m resolution was derived based on SDGSAT-1 TIS in three bands."

How all three bands were used together in the lead detection? You must explain this in detail.

**4. Results**

l. 302: spell out TP, FP, FN and TN in the text,

Section 4.2: equation numbers should be (2) and (3).

Section 4.3: again how lead detections by all three TIS bands are combined?

l. 381: "The previously developed method (Qu et al., 2021) was applied to detect the leads based on the MODIS IST data."

Explain in the text why you selected the method by (Qu et al., 2021).

**5. Discussion**

You talk in the text about 'ozone resolution', what is that? Figure 12 shows 'best total ozone solution'.

You could add to the discussion here the correlations between different TIS bands.

Section 5.2: Descriptions on S-1 SAR processing should go to Data Section.

l. 483: "In particular, the B2 band is more sensitive to such surface information because various types of sea ice have different emissivity and produce different BT values."

Are the TIR emissivities of different sea ice types really that different? Please check literature, and give variation range in the text. I would say it is more about variation of sea ice/snow surface temperature, e.g. thin sea ice has surface temperature which is a function of freezing temperature and air temperature, but surface of thick ice is fully insulated from ice bottom, i.e. only air temperature matters.

l. 498: "On the other hand, as the TIS data available within the scope of this paper is relatively limited, these individual case studies presented may be weak in terms of generalizability."

Yes, this is the case, and this must be also emphasized in Conclusion Section.

**Technical corrections**

Many figures are too small, try to increase their sizes. Also colorbars for BT, BTA, are way too small.

References for the same authors and the same year are missing a,b,c, after the year in Reference list.

---

## Community Comment (CC1)

**Response on RC1**

We thank the reviewer for these helpful comments and suggestions. We have addressed the comments and will implement the proposed changes in the manuscript. Please, see below our detailed response. Major Comments were responded to in paragraphs, and Line Comments were responded to point by point.

Our response is highlighted in blue. We will provide a track-change version together with the revised paper.

**Summary**

The article provides an overview of a new satellite that allows for detection of sea ice leads at a spatial resolution not possible with any other satellite. This is novel and interesting, however, the results as presented appear to over-emphasize the accuracy and precision of the results given the limited extent of the analysis.

**Response:** Thanks for your time and constructive comments on our manuscript. We have addressed the limitations of this study in Conclusion section:

"Nevertheless, limited by the imaging time and cloudy conditions over the Arctic region, only individual case studies based on TIS data were carried out."

**Major Comments**

The comparisons against other moderate resolution products are incomplete. The focus of the analysis presented is on narrow leads that go undetected at moderate resolution (1km). The equally important but overlooked analysis would be the moderate (1km) resolutions results interpolated into higher resolution (30m). When detecting the same lead, how often do moderate resolution results over estimate (or under estimate) the lead area relative to the higher resolution detections. The claim that high resolution results detect more lead area because it can detect narrower leads is incomplete without establishing that moderate resolution lead detections do not have a bias in lead area. Because, visual inspection of the results suggest that

moderate resolution lead detections have a bias in terms of over-representing lead area for leads that are wide enough to be detected. It may be true that more leads can be detected by a 30 m resolution imager, but the total lead area detected at 30 m resolution may not necessarily be higher overall if the 30 m lead detections do not have the same bias towards over-representing lead area for larger leads.

**Response:** In fact, we did not intentionally interpolate the MOIDS-derived leads to 30 m. The sea ice lead area derived from MODIS IST in Table 6 was calculated from at its original resolution of 1 km, while the area derived from the TIS was calculated at a resolution of 30 m.

We tried to understand that the argument raised by the reviewer, which might be briefly summarized "more leads detected by TIS does not mean that more lead areas obtained compared to the MODIS data". This might be true. However, interpolation of the coarse-resolution (1000 m) to a high-resolution (30 m) probably can induce more problems on uncertainty of the brightness temperature data than the detection results using the original data. This is beyond scope of the presented study. Some previous studies applied interpolation methods to obtain high-resolution lead detection, and then estimated heat fluxes over them. For example, Qu et al. (2019) used cubic convolution interpolation to resample the Landsat-8 TIRS imagery from the resolution of 100 m to 30 m. Their results showed that the high-resolution TIRS data gave a slightly smaller lead area but a larger lead length compared to the MODIS data, which resulted in small leads accounting for more than a quarter of total heat flux. Applying a convolutional neural network, Yin et al. (2021) obtained the super-resolution MODIS data at 100 m resolution and more reliable heat flux estimations than those at original 1 km resolution and interpolation-based high resolution. But as we explained above, interpolation of the remote sensing data from coarse resolution to high resolution can induce some uncertainties, particularly for this study that TIS data has a resolution of 30 times better than the MODIS data.

Although we cannot assess the extent to which moderate-resolution detection under or over estimates results compared to high resolution, the statistical results

shows that the lead area derived by the TIS at high resolution is larger than the moderate-resolution result by the MODIS IST. Thus, the leads that were unobserved at moderate resolution contribute more to the detection result in the cases of this study.

The justification for using a 3 channel approach is a little weak. There is mention of how there is ozone absorption in band 1, but the authors appear to be correlating ozone absorption with the air temperature near the surface rather than the air temperature in the ozone layer in the upper atmosphere. Also, the analysis of how the 11 and 12 micron channels (bands 2 and 3) compare in their sensitivity to ice vs water clouds is lacking; again the thermal contribution of water vapor and ice would tend to be much higher in the atmosphere than 2m.

**Response:** The SDGSAT-1 TIS has three infrared bands, all of which play an important role in surface temperature retrieval. Apart from the B2 and B3 bands, the B1 band is also a valid infrared channel for temperature measurement that can provide additional thermal information for use with B2 and B3 in the triple split window algorithm (Liu et al., 2021). The detection accuracy of the B1 band is slightly lower than the B2 and B3 bands, but still within a satisfactory level. We agree that for robustness purposes, it would be better to use the infrared band with the best detection performance (e.g., the B2 band). However, as the first study to apply the SDGSAT-1 TIS for lead detection, it is necessary to demonstrate the application of a combination of the three bands in addition to individual bands in this study.

With respect for the ozone data used in the Discussion section, we aimed to analyze the sensitivity of the B1 band to ozone solution, which were pointed out by previous studies (Wan and Li, 1997). Our results show that although ozone affects the absolute temperature of the B1 band, it does not diminish the thermal contrast, i.e., it does not affect the performance of lead detection.

It is true that the sensitivity of the B2 and B3 bands to ice and water clouds has not been analyzed in this study, because the low cloud cover in the selected TIS data

makes such a comparison unnecessary.

Indeed, the correlation analyze between temperature characteristics and total cloud cover and total column water vapour (based on ERA5 product) has been carried out, but no correlation was found. It may be that the spatial resolutions (30 m for TIS vs 0.25 degree for ERA5) differ so much, or as mentioned earlier the TIS data selected are relatively clear, that it is difficult to detect correlations.

**Line Comments**

Line 9: Use "sea ice" rather than "sea ice cover".

**Response:** Done as it is suggested.

Line 9: Use "between" rather than "from" because the heat exchange can go in either direction.

**Response:** Done as it is suggested.

Line 15: The I-band on VIIRS has an IR resolution of 375m at 11 microns. So I would say that the resolution is an order of magnitude improvement rather than 2 orders of magnitude; hundreds of meters rather than kilometers.

**Response:** Thanks for pointing it out. We modified the sentence to "the spatial resolution of leads by infrared remote sensing increases from the scale of hundreds to tens of meters" in the revised version.

Line 16: It does not seem appropriate to attribute 4 significant digits to the results; the results may be 96% accurate, but the precision of the 0.01% is not likely.

**Response:** Thanks for your comment. After further consideration, we decided to use three significant digits, such as 96.3%. The corresponding tables and descriptions have been revised in the revised version.

Line 35-36: There are examples of the heat flux occurring in either direction (see above comment about line 9)

**Response:** Done as it is suggested.

Line 54: This overlooks the importance of clouds, cloud shadows, and the thermal contrast as ice ages.

**Response:** We agree with you that it is important to discriminate leads from clouds and cloud shadows and different ages of ice. The sentence has been modified:

"For sea ice lead detection based on thermal infrared data, the key lies in deriving thermal contrasts, namely, the temperature anomaly between sea ice and open water, and to distinguish leads from thermal contrasts caused by ice ages and clouds" in the revised version.

Line 76: This 1993 paper is too old – it references AVHRR – which does not describe the detection capabilities of modern satellite imagers.

**Response:** Yes, it is true that AVHRR was used in (Key et al., 1993). However, we cited it here because it suggested the effect of sensor resolution on lead statistics and therefore highlighted the importance of observing leads in a fine scale. We have replaced it with another literature, but also old. Since the effect of sensor resolution on lead statistics is well known, there has been relatively little research in recent years.

The corresponding sentence has been modified as: "Key et al. (1994) assessed the effect of sensor resolution on lead width statistics. They suggested that the mean lead width 'grows' as the pixel size builds up in gradually degraded images."

Line 78: Clarify what is meant by "resolve". Satellite imager can "detect" sub-resolution thermal emissions – if the thermal contrast is larger enough. Does this line mean just mean that it hard to attribute a width to sub-resolution features?

**Response:** We agree that for leads with widths less than pixel resolution, it is indeed possible to be represented as "mixed pixels" in thermal images. The sentence was deleted.

Line 92: The word "parallel" can be removed, it does not add any descriptive value.

**Response:** Done as it is suggested.

Line 111, Table 1: Is the expected noise still on the order of 0.2K at the cold end of the temperature range?

**Response:** The onboard blackbody is the primary calibration source for SDGSAT-1 TIS, and usually controlled at a constant low temperature. In the prelaunch test (Hu et al., 2022), the blackbody temperature varies from 240 K to 300 K. The NEdT at 300 K for the three TIS bands is 0.034 K, 0.047 K and 0.076 K respectively, which satisfy the pre-flight requirements (<0.2 K). Noise rises mainly with increasing temperature. Thus, although no NEdT was recorded at the cold end, the NEdT at nominal blackbody background temperature of 300 K is essentially representative of the performance of the TIS. The corresponding reference is:

Hu, Z., Zhu, M., Wang, Q., Su, X., and Chen, F.: SDGSAT-1 TIS Prelaunch Radiometric Calibration and Performance, Remote Sensing, 14, 4543, 10.3390/rs14184543, 2022.

Line 131: What is level-4 data?

**Response:** The SDGSAT-1 data products include different Level-1, Level-2 and Level-4 data products. Level-1 product is a standard product based on the Level-0 product, after data processing such as relative radiometric correction, band registration, HDR fusion, etc. Level-2 product is based on the Level-1 product after geometric correction. Level-4 product is based on the Level-1 product after ortho-rectification using ground control points and Digital Elevation Model (DEM) and output with standardized format. Currently, only Level-4 product is available to users. We have added corresponding reference in the revised version:

International Research Center of Big Data for Sustainable Development Goals (CBAS): SDGSAT-1 Data Users Handbook (Draft), 2022. http://60.245.209.56/preview/20221125/c84c0b5d89984cd384ffa05dbb163d14.pdf

Line 135: The first sentence is hard to understand. Could it be rephrased as "is a two satellite constellation" rather than "is formed by two satellites"?

**Response:** The sentence is modified as "Sentinel-2 (S2) is a constellation of two satellites, S2A and S2B".

Line 139: Do you just mean that 560 nm is close to green on the visible spectrum? Why is that important? Is this saying that leads appear to be green in color?

**Response:** We visually compared S2 visible images (bands 2, 3 and 4) and found good discrimination between leads and surrounding sea ice from green band images. This may be because the green band primarily reflecting the differences between leads and sea ice rather than the differences between different types of sea ice, due to the generally low top-of-atmosphere (TOA) reflectance of sea ice in the green band (König et al., 2019). Therefore, it is appropriate to use the S2 green band images as the reference for validation. However, it does not mean that sea ice leads appear to be green in color.

Analogous situation can be found in the reflectance histogram of the three S2 bands (see Figure R1), i. e., the reflectance of the green curve is lower than the other two. For each histogram curve, the highest peak represents the surrounding sea ice, while the gentle slope and the lower peak represent the lead with seawater and thin ice. We noted that both peaks in the green band are more prominent, producing a good discrimination between sea ice and leads. It is easier to select a threshold at the valley between the two peaks (after obtaining the normalized image).

König, M., Hieronymi, M., and Oppelt, N.: Application of sentinel-2 MSI in Arctic research: Evaluating the performance of atmospheric correction approaches over Arctic sea ice, Frontiers in Earth Science, 7, 22, 10.3389/feart.2019.00022, 2019.

[Figure]

Figure R1 Histogram of reflectance in S2 band 2 (red line), band 3 (green line) and band 4 (blue line).

Line 150 & 151: Remove "The" in front of "MODIS".

**Response:** Done.

Line 152: Why are level 3 products used instead of level 2 (or level 1)?

**Response:** Thanks for your valuable comment. We have conducted the comparison experiment for level-2 MOD29 and SDGSASAT-1 TIS data. Corresponding MOD03 geographic products were also used, as the data information listed in Table R1.

Leads were detected from the MOD29 data by the same method described in the original manuscript. The statistics of the lead area based on different data are listed in Table R2. Overall, the area estimated from the TIS data is larger than that estimated from MODIS IST, with the total area exceeding the latter by a third. The second comparison (on April 28) shows more reasonable result. These experiments will be presented in the revised version (if the editor decides the manuscript can be revised). Correspondingly, the correlation experiment in section 5.1 will be modified.

We would like to re-emphasize that while moderate resolution sensors over-represent the width of leads, this study showed that the narrow leads overlooked at moderate resolution are more important and contribute more to the overall lead area.

Table R1 Information of MODIS products used in this study

|  |  | MOD29 | MOD03 |
|---|---|---|---|
| Date and time (UTC) | 2022-03-23 | 10:30 | 10:30 |
|  |  | 12:05 | 12:05 |
|  | 2022-04-03 | 05:10 | 05:10 |
|  | 2022-04-28 | 05:05 | 05:05 |
| Spatial resolution |  | 1 km | 1 km |

[Figure]

(a)                        (b)                        (c)

Figure R2 Comparisons between lead detection based on MOD29 and SDGSAT-1 TIS data in the Beaufort Sea on April 3, 2022. (a) Level-2 MODIS IST products, where the clouds are off-white, the land is dark gray, and the overlaid black border denotes a coverage for (b), (c). (b) Leads at 1 km resolution derived by MODIS IST product. (c) Leads at 30 m resolution derived from the combined result of SDGSAT-1 TIS B1, B2 and B3 bands.

Table R2 Statistics of the lead area estimated from the MODIS IST data (level 2) and the SDGSAT-1 TIS data.

|  |  | Sea ice lead area (km²) | | | Additional lead area by the TIS than by Hoffman et al. (2022b) (km²) |
|---|---|---|---|---|---|
|  |  | MODIS IST | SDGSAT-1 TIS | TIS/MODIS |  |
| 1 | Beaufort Sea on April 3 | 14,283 | 15,362 | 1.08 | 5,679 |
| 2 | Beaufort Sea on April 28 | 4,238 | 10,500 | 2.48 | 4,590 |
| 3 | Laptev Sea on March 23 | 4,021 | 4,519 | 1.12 | 1,462 |
| 4 | Laptev Sea on March 23 | 3,886 | 3,936 | 1.01 | 2,415 |
| | **Total** | **26,427** | **34,318** | **1.30** | **14,145** |

Line 158: While it is true that MODIS-Terra crosses the equator in the morning, this does not have any correlation with what time of day the satellite will provide coverage in the Arctic. And again, level 1 or 2 products will provide a better time-match than averaged level-3 products.

**Response:** Thanks for your comment. MODIS-Terra based products can be matched to SDGSAT-1 well, with an average time difference of 1 hour. Therefore, we mainly used the MOD29 IST data. We have taken your advice and conducted the comparison experiment for level-2 MOD29 and SDGSASAT-1 TIS data. Please refer to the previous response for the comparison.

Line 160-164: What is the importance of the near-surface air temperature? If there is ozone, water vapor, or ice crystal absorption, those phenomena would be occurring much higher in the atmosphere.

**Response:** Thanks for your comment. We agree with you that the absorption effect of atmospheric components would be significant for temperature dataset without atmospheric effects removal. However, we focused the discussion on the temperature characteristics of leads. There is a physical dependence between the evolution of the leads (or sea ice) and the near surface air temperature.

On the one hand, as leads would refreeze quickly at low air temperature (e.g., in the winter), changes in air temperature are important to leads. On the other hand, leads allow for strong heat exchange between the ocean and under atmosphere. Opening leads may potentially change air temperature (Lüpkes et al., 2008). Furthermore, near-surface temperatures and IST are often correlated (such as a spatial correlation shown in Fig. 12 (a) and (b)). Please refer to the corresponding literature:

Nielsen-Englyst, P., Høyer, J. L., Madsen, K. S., Tonboe, R., Dybkjær, G., and Alerskans, E.: In situ observed relationships between snow and ice surface skin temperatures and 2 m air temperatures in the Arctic, The Cryosphere, 13, 1005-1024, 2019.

Line 170-174: Without objecting to the accuracy of this section, I don't know that it is necessary for this paper.

**Response:** Please refer to the previous response.

Figure 2: The chart shows 1 path going to Step 1 and 1 path going straight to Step 2. By what logic can Step 1 be bypassed?

**Response:** Step1 is not bypassed by Step2. After Step1, Step2 uses two input

data. One of them is the original BT data (which was also the input to Step1); the other is the potential leads output from Step1.

We have amended the corresponding descriptions in the revised version: "the first step of our lead detection is applying a binary segmentation to extract potential leads from the BTA data. In the second step, the potential leads are used together with BT data in a designed filter to obtain the consequent leads".

Line 228: On example is presented, is this representative of what other cases look like?

**Response:** There are prevalently different effects when selecting three different thresholds of 1.2, 1.5 and 2.7 K base on the TIS data used in this study. We have recognized that the result is not representative of other seasons and seas since the TIS data are limited. However, the case shown here is quite representative with both clear large and narrow leads, as well as wealth of temperature variations. Based on this typical example, we aimed to show each step result of the proposed detection method. In the Result section of the manuscript, we have presented few individual cases of the lead detection from the Beaufort Sea in April 2022, and also included a complex scenario from the Laptev Sea in March 2022, all of which demonstrated good applicability.

Line 229: False-positive detections could be clouds, cloud shadows, or cloud edges; not just sea ice.

**Response:** Thanks for pointing it out. We modified the sentence as "false-positive detections (i.e., sea ice or others classified as leads), e.g., the white pixels marked by the orange square in Fig. 5 (a)"

Line 232: "Multiple tests" needs further explanation.

**Response:** We tested the BTA threshold segmentation when the BTA threshold was varied from 1.2 K to 2.7 K in steps of 0.1 K and visually compared their segmentation effects. The corresponding sentences are modified:

"Multiple threshold segmentation was tested when the BTA threshold was varied from 1.2 K to 2.7 K in 0.1 K steps. After visual comparison, using 1.8 K threshold resulted in a significantly different segmentation effect from the previous step, with minimal differences from the next step.".

Figure 7: Do not slit the figure across 2 pages. And, in (a), what do the numbers 1,2,3 mean?

**Response:** Thanks for pointing it out. We will improve these figures in the revised manuscript. We have explained the meaning of the numbers 1,2,3 in Figure 7's caption: "These thresholds are 1: mean plus standard deviation (std) of BT before segmentation, 2: iterative threshold, and 3: Otsu's threshold."

Line 247: Hard to follow.

**Response:** The sentence is revised as: "False positive detections can be attributed to imperfectly removed clouds, cloud edges, or sea ice of different thicknesses. These interferences cause gradient variations in the BT values measured by the TIS sensor yielding high BTA values."

Line 248: Which temperature gradient are you talking about? Ice surface temperature, retrieved brightness temperature, surface air temperature?

**Response:** We meant the gradient of the TIS brightness temperature. The sentence is modified in the revised version. Please refer to the revised statement in the previous response.

Line 253: The false positives are likely clouds or cloud artifacts.

**Response:** We agree that false positives are likely clouds or artifacts. In general, clouds have low temperature but high BTA values in the edges. We were careful to mention this in the revised version. Please refer to the revised statement in the previous response.

Line 256: Does "in view 1" mean Figure 7, Panel 1?

**Response:** Yes. Both "View 1" and "View 2" are taken from the left-hand panel in Figure 7. Their corresponding BTA images are shown in the first row on the right parallel panels, and the BT images are shown in the last row. The corresponding sentence is amended "in the second row of right parallel panels, the absolute values of the BT of those reliable leads in the first column (in view 1) are all greater than those of the false-positive detections in the second column (in view 2) by at least 2 K."

Line 257: Is this true for just this case, is it also true for other times of day or times of year when the ice and clouds may tend to have different temperatures?

**Response:** The distinguishability between leads and false-positive detections (i.e., false positives generally have lower BT values than highly reliable leads, despite having high BTA values) was found in all the 11 TIS data used in this study. Since the SDGSAT-1 data over the Arctic are still ongoing collected, we have not studied cases in other seasons, but this will be our future research focus.

Line 258: When you say "remaining", where do you mean, Figure 7 (a)?

**Response:** We have recognized that this can been misleading and have revised it to "The BT histogram for those potential leads is shown in Fig. 7 (a)".

Line 279: "co-located" is more commonly used than "collocated".

**Response:** Done as it is suggested.

Line 281: Is this just saying that leads are darker than ice in the visible spectrum? I think it is already well understood that water is visually darker than ice.

**Response:** Thanks for pointing it out. We will simplify the sentence in the revised manuscript: "For the matched visible images, leads are darker than ice surface".

Line 285: So this is not a binary mask, it could have 3 outcomes: lead, non-lead, ambiguous?

**Response:** Yes, the normalized brightness can be interpreted as three cases:

1) A pixel with normalized brightness between 0 and 0.07 must be a lead;

2) a pixel between 0.07 and 0.7 is an ambiguous case;

3) a pixel between 0.7 and 1 must be sea ice.

However, in order to compare with binary lead detection, the second ambiguous case was subsumed into the "possible lead" and "possible sea ice". That is, both the first and second cases are "possible lead"; both the second and third cases are "possible sea ice". As what we mentioned in the original manuscript: "a pixel with a normalized brightness below 0.7 could be a lead, while a pixel with a normalized brightness above 0.07 could be sea ice."

Figure 8: Why is the B1 brightness temperature (blue) so much colder than B2 & B3 (yellow)?

**Response:** The B1 band is centered at 9.3 μm (8.0-10.5 μm). Comparing to the B2 and B3 bands, its radiation is attenuated by ozone. The previous studies (Wan and Li, 1997; Prabhakara and Dalu, 1976) also demonstrated that the 9.3 μm channel has colder brightness temperature. This is exactly the reason we decided to investigate the influence of ozone on the lead detection by the TIS B1 band and the correlation between ozone solution and the three TIS bands. Thus, we would like to remain the corresponding content in sub-sections 2.5 and 5.1.

Table 3: How can you have a binary comparison when you are excluding brightness values between 0.007-0.7 (see not on Line 285)?

**Response:** In this study, a pixel with a normalized brightness below 0.7 could be a lead (< 0.7), while a pixel with a normalized brightness above 0.07 could be sea ice (> 0.07). We did not exclude the overlapping interval between these two cases (0.07 - 0.7), but set the interval as an ambiguous case, as shown in Figure R3.

[Figure]

Figure R3 Diagram of a binary cases (sea ice and lead) for the normalized brightness.

Table 5: How do these numbers change if the temperature threshold is changed?

**Response:** Thank you for your question. The method proposed in this study used two thresholds. One of the two is the BTA threshold of 1.8 K. The other is the BT threshold selected by an iterative method based on the previous segmentation result (potential leads). The BT threshold is not fixed and generally varies with the temperature characteristics of the potential leads. Thus, the statistics of the lead detection for the TIS three bands shown in the Table 5 of the manuscript are less dependent on the temperature threshold. In contrast, the BTA threshold is often a fixed constant. For example, Hoffman et al. (2019) identified a threshold of 1.5 K; Qu et al. (2021) took 1.2 K, 1.5 K and 2 K as thresholds for different types of leads, corresponding to large to small uncertainty levels.

Line 360: Plead define strip noise.

**Response:** Strip noise is a sharp fluctuation in DN values that occurs when imaging a homogeneous surface due to different noise bias given by each detector. This results in a strip of noise in the scanning direction of the TIS sensor. We have mentioned this in sub-section 2.2: "The B1 band shows less strip noise (i.e., signal fluctuations along the sensor scan caused by detector noise) than the other two bands.".

Table 6: Do not split table across two pages. Also, the ratio is 3 cases are on the order of 1, but Case 2 is more than 5 times higher. It is hard to generalize a relationship with such a big outlier yet a small sample size.

**Response:** Thanks for your valuable comment. In the added comparison experiment using the level-2 MODIS IST product, Case 2 yields a ratio of 2.48 (see

the Table R2), which is more reasonable than before. The different statistics between Case 2 from others may be due to significant changes in leads in the late spring (near the melting season).

For a more general comparison, we have collected another lead dataset based on MODIS data (Hoffman et al., 2022b) and conducted a comparison. The result was listed in the last column of Table 6 in the original manuscript. More details in this comparison are presented in the following Table R3. Please note the difference between the two "All lead areas" and the "Additional lead area". The latter is not a direct subtraction between the first two datasets, but rather the redundant area after a mask processing (as we have mentioned in the original manuscript). Here, the result of Case 2 is reasonable, in which the additional lead area detected by the TIS than by Hoffman are 4,590 km$^2$.

Table R3 Statistics of the lead area estimated from the lead dataset of Hoffman et al. (2022b), referred to as Hoffman in the following, and the SDGSAT-1 TIS data.

| | | All lead area (km$^2$) | | Additional lead area by the TIS than by Hoffman (km$^2$) |
| --- | --- | --- | --- | --- |
| | | Hoffman | SDGSAT-1 TIS | |
| 1 | Beaufort Sea on April 3 | 34,170 | 15,362 | 5,679 |
| 2 | Beaufort Sea on April 28 | 20,287 | 10,500 | 4,590 |
| 3 | Laptev Sea on March 23 | 14,566 | 4,519 | 1,462 |
| 4 | Laptev Sea on March 23 | 4,109 | 3,936 | 2,415 |
| | **Total** | **73,132** | **34,318** | **14,145** |

Line 441: Do you mean 30 m instead of 30 km?

**Response:** The number is right. We extracted the BT and BTA data only for the lead pixels and allocated them to the geographic grids at 30 km, i.e., one tenth of the TIS swath width, so that we can easily compare the temperature characteristics with the coarse-resolution datasets (ERA5 air temperature data and OMI/Aura product in a

regular grid of 0.25 degrees).

Figure 12: Why is the 2m air temperature shown? If Ozone is contributing to the brightness temperature retrieval, that thermal contribution would be coming from much higher in the atmosphere.

**Response:** Same as our previous response, the reason we used near-surface temperatures is to investigate the correlation between leads and air temperatures. There were some studies exploring the relationships (Qu et al., 2021; Qu et al., 2019; Nielsen-Englyst et al., 2019; Yin et al., 2021). We were not suggesting that near-surface temperatures are related to ozone.

Nielsen-Englyst, P., Høyer, J. L., Madsen, K. S., Tonboe, R., Dybkjær, G., and Alerskans, E.: In situ observed relationships between snow and ice surface skin temperatures and 2 m air temperatures in the Arctic, The Cryosphere, 13, 1005-1024, 2019.

---

## Community Comment (CC2)

**Response on RC2**

Thanks for your time and helpful feedbacks on our manuscript. We have addressed the comments and will implement the proposed changes in the manuscript. Please, see below our detailed response. General comments were responded to in paragraphs, and Specific Comments were responded to point by point.

Our response is highlighted in blue. We will provide a track-change version together with the revised paper.

**General comments**

The authors have developed a method to map Arctic sea ice leads using Thermal Infrared Spectrometer (TIS) data onboard recently (Nov 2021) launched Chinese Sustainable Development Science Satellite 1 (SDGSAT- 1). TIS has three IR bands: B1 (8.0- 10.5 μm), B2 (10.3- 11.3 μm) and B3 (11.5- 12.5 μm), with 30 m resolution and 300 km swath width. The TIS instrument provides TIR data at much finer resolution than MODIS (1 km) and VIIRS (750 m), and somewhat finer than Landsat 8/9 (100 m). Optical and NIR data at comparable or even finer resolution are available from many sensors (e.g. Sentinel-2, Landsat 8/9), but only TIR data enables to retrieve sea ice properties regardless of daylight conditions.

The method for the Arctic sea ice leads mapping is developed using 11 TIS images acquired over the Laptev and Beaufort Seas in Match and April 2022. The general lead detection method was same for all three TIS bands with some variable parametrization, i.e. all three bands were not used together for the lead detection, but in some cases the resulting three lead maps were combined (if I understood correctly, please see my comment later). The same TIS images were also processed to the leads maps (binary map: lead (open water or thin ice) or sea ice. The leads by the three TIS bands were compared to each other, and to lead detection maps by Sentinel-2 band 2 (green band) data and MODIS daily IST data. The Sentinel-2 lead detection was based on study by Muchow et al. (20121) and the MODIS lead detection to (Qu et al.

2021). A case study was also conducted were a TIS lead map was compared to Sentinel- 1 dual-polarization SAR image. Possible atmospheric influences to the TIS lead detection were investigated using ERA5 air temperature and OMI ozone total column data. The TIS band 1 covers the ozone absorption band.

In general, the TIS lead detection seemed work fine, and against the S-2 lead maps there was 96.3% pixel based match (authors used TIS accuracy here). Compared with the MODIS lead map, the TIS map presents more leads with width less than hundreds of meters. All three TIS bands showed similar performances in detecting leads. The B1 band can be complementary to the other two bands, as the temperature sensitivity is different from the other two, benefiting better detection by combining the three bands. Arctic lead maps with fine resolution (e.g. 30 m) allows to observe narrow leads which are undetected in the MODIS/VIIRS products, and estimate their contribution to the overall Arctic lead fraction.

I think the study set up with data acquisitions and data processing is rather well conducted, and the TIS lead detection method could be generally applicable for a large number of TIS images, but I think it is not sure as it is based on small amount of data. Further, I don't think it is meaningful to develop lead detection methods for the three TIS bands separately, and compare the results. You should develop the best possible lead detection algorithm for the TIS data (having as input all bands or just two/one), and only present this in the paper. In the following I have also some other major comments to the papers and suggestions for possible improvements. These are followed by miscellaneous specific ones.

**Response:** Thanks for your valuable comment. We agree that it would be better to apply the most appropriate thermal infrared band for lead detection. However, until the cross-comparison experiments were carried out, we found each band has its advantage on ice leads detection.

The TIS three thermal infrared bands showed different radiometric accuracy during the commissioning phase. The absolute radiometric calibration evaluation by

Hu et al. (2022) suggested that the average temperature bias of SDGSAT-1 TIS reached 0.661 K, 1.081 K and 0.426 K for B1, B2 and B3, respectively. This suggests that the B3 band has the best radiometric calibration accuracy. However, B2 and B3 bands are more affected by the strip noise than the B1 band. B1 band is a less common thermal infrared channel with colder temperatures than the other two split-window channels (B2 and B3 bands) due to the absorption effect of ozone. Few studies have investigated its similarities and differences with the 11 and 12 μm TIR channel for sea ice remote sensing. However, we do find that using the TIS B1 band can obtain more small leads in the presence of interference in B2 and B3 data.

Taken together, it is necessary to carry out a comprehensive application demonstration of the new on-board sensor. We applied the same method to the three bands of SDGSAT-1 TIS to extract leads, and further conducted cross-comparison to determine their detection performance. The cross-comparison results suggest it is beneficial to combine the lead detection results of the TIS three bands.

The review and discussion on related previous studies is well conducted, e.g. different methods for lead detection are nicely discussed. However, your review could include also following new study:

Q. Wang, M. Shokr, S. Chen, Z. Zheng, X. Cheng and F. Hui, "Winter Sea-Ice Lead Detection in Arctic Using FY-3D MERSI-II Data," in *IEEE Geoscience and Remote Sensing Letters*, vol. 19, pp. 1-5, 2022, Art no. 7005105, doi: 10. 1109/LGRS.2022.3223689.

There are many studies with lead detection using SAR and altimeter data, and it is fine to give only few examples as you have done. Related to (Murashkin and Spreen, 2019) include also reference:

Dmitrii Murashkin, Gunnar Spreen, Marcus Huntemann, and Wolfgang Dierking, "Method for detection of leads from Sentinel- 1 SAR images," Annals of Glaciology, vol. 59, no. 76, pp. 124– 136, 2018

Could you use their method for automatic lead detection in your S-1 SAR imagery? This would allow better utilization of the SAR imagery as comparison data.

Your review includes only one study where Landsat data are used for the lead

detection (Qu et al. 2019). Are there any other Landsat studies? Please check. One relevant study here could be:

Cáceres, A.; Schwarz, E.; Aldenhoff,W. Landsat-8 Sea Ice Classification Using Deep Neural Networks. Remote Sens. 2022, 14, 1975. https://doi.org/10.3390/rs14091975

Sentinel-2 lead study by (Muchow et al. 2022) must be discussed in Introduction, it is now mentioned later in Section 4.1.

You could also summarize currently publicly available Arctic lead products with their time spans, seasonal coverages (full year or only winter season) and spatial resolutions.

**Response:** Thank you very much for your advice. We carefully checked relevant literatures and restructured the corresponding paragraphs in the Introduction section to better review recent studies. Table R1 shows the current publicly available Arctic lead datasets developed by different methods at different resolution with time spans, which will be added to the revised version.

It should be noted that the sea ice classification algorithm developed by Cáceres et al. (2022) based on Landsat-8 focuses on the different sea ice types (ice free, gray to white ice, thin first year ice and medium first year ice) in the Baltic Sea. No lead detection or observation is involved. Therefore, this paper is not cited.

In Section 5.2, the purpose of the comprehensive analysis incorporating Sentinel-1 data with the TIS detected results is to explore the property within the lead. For this purpose, we have specifically analyzed a complex scenario congaing a potential transition zone between thin ice and seawater, as shown in Figure 13 in the original manuscript. Although we did not use the automated S1 lead detection algorithm (Murashkin and Spreen, 2018), the S1 images in dual-polarization provide valuable backscatter information than a binary result. The backscatter of the lead transition zone, which is higher in the S1 HH image and lower in the HV image, was consistent with the "bright lead" feature described by Murashkin and Spreen (2018). However, the backscatter of surrounding ice is rather inhomogeneous and the contrast

with the lead is not significant in both the HH and HV images. Notably, Murashkin and Spreen (2018) did not analyze this situation. Therefore, the automated lead detection algorithm may not be adapted to the scenario we have shown here. In contrast, it is more appropriate to analyze the differences between HH and HV data directly, so we have performed a false-color composite using the dual-polarized data.

Table R1 Arctic lead datasets and with their spatial resolution and time span.

| Dataset | Satellite sensor | Spatial resolution | Time span | and seasonal coverage |
|---|---|---|---|---|
| Bröhan and Kaleschke (2014) | AMSR-E | 6.25 km × 6.25 km | 2002 to 2011 | November to April |
| Willmes and Heinemann (2015b) | MODIS | 2 km$^2$ | 2003 to 2015 | January to April |
| Reiser et al. (2020) | MODIS | 1 km$^2$ | 2002 to 2021 | November to April |
| Hoffman et al. (2021) | MODIS | 1 km$^2$ | 2002 to 2022 | November to April |
| | VIIRS | 1 km$^2$ | 2011 to 2022 | November to April |

Section 2 Data could be changed to "Data and pre-processing", i.e. to include all data processings before analyses and lead detection method development, e.g. move Section 3.1 (Pre-processing of TIS data) to Section 2.

Detailed descriptions of TIS instrument and its data should moved from Introduction to Section 2.1. In the following are some questions on the TIS data:

What is the size of the TIS image along track?

What is the main intended application of the band 1?

What is the resolution of the TIS data in K?

Is there yet IST product from bands 2 and 3? Under development?

Are there any TIS cloud masking algorithms or products?

**Response:** In accordance with your suggestions, we amended the data presentation and pre-processing sections (please refer to the revised version if the editor decides the manuscript can be revised). We would like to answer your questions about the TIS data here (and have added these details where appropriate in the revised manuscript).

- For convenient use of the TIS data, the ground segment crops the original TIS data to 300 km in the along-track dimension.

- TIS B1 band is a wide channel with a wavelength of 8.0-10.5 μm. It is mainly used in combination with the B2 (10.3-11.3 μm) and B3 (11.5-12.5 μm) bands to obtain a better accuracy in land surface temperature retrieval based on the three-channel split-window algorithm (Liu et al., 2021; Hu et al., 2022).

  Hu, Z., Zhu, M., Wang, Q., Su, X., and Chen, F.: SDGSAT-1 TIS Prelaunch Radiometric Calibration and Performance, Remote Sensing, 14, 4543, 10.3390/rs14184543, 2022.

- Strip noise is a sharp fluctuation in signals that occurs when imaging a homogeneous surface due to different noise bias given by each detector. In general, the TIS B1 band has relatively less strip noise. We have added this to Section 2.2.

- The quantization bit of the TIS is 12 bit. The TIS radiometric measurement is better than 0.42 K for the three bands, which satisfies the preflight requirements (≤1 K)

- There are currently no TIS-based surface temperature products or cloud mask products available, all of which are under development.

BTA threshold for the lead detection was manually selected (here 1.8 K). Are you sure this is really applicable for a large amount of TIS images acquired in various sea ice and atmospheric conditions? Why did you not develop an automatic selection method for the BTA threshold, as you did for the BT threshold? I think is rather serious potential flaw in your lead detection method. You should really have an automatic BTA threshold selection method.

**Response:** We did consider using iterative thresholds for the BTA data as well, as Willmes and Heinemann (2015a) did. However, it is hard to argue that automatically selected thresholds are more appropriate than fixed thresholds for few cases in this study. For the three bands lead detection, without the use of a fixed BTA threshold for standard, the comparability of binary segmentation results would be poor, and the further cross-comparisons between the three results would be meaningless.

Although the TIS data used in this study cannot yet include various sea ice and atmospheric conditions, we would like to explain here the soundness of the constant threshold. We tested the results of the threshold values selected by the iterative method. Setting the initial threshold as 1.2 K, the automatically selected BTA thresholds by iterative method for the seven TIS data (for the each of the three bands) were shown in Table R2. The iteration thresholds for the BTA images were relatively close, with the minimum of 1.8 K and the mean of 2.0 K. From this perspective, no large errors can be produced between the segmentation results from iterative selection or from the constant threshold.

Table R2 BTA iteration thresholds for three bands based on seven TIS data

| | | | | | | | |
|---|---|---|---|---|---|---|---|
| **B1** | 1.9 | 2.2 | 1.9 | 2.2 | 2.0 | 1.9 | 2.5 |
| **B2** | 1.8 | 2.1 | 2.1 | 2.0 | 2.0 | 2.0 | 1.8 |
| **B3** | 1.8 | 2.1 | 2.0 | 1.9 | 1.9 | 2.0 | 2.4 |

In general, we could have better reliability on your TIS lead detection method if more TIS data had been used in its development. Can you add more TIS images to your study?

**Response:** Although we would like to carry out more detections, what we have presented in this manuscript is all that can be done in the spring of 2022. On the one hand, the SDGSAT-1 was launched just one year ago, so we can only obtain data after March 2022. On the other hand, the cloud interference is the main limitation for lead

detection based on thermal infrared data in the Arctic. Due to the unavailability of simultaneous cloud detection (we are also working on this point), the method proposed in this study is only concerned with clear sky conditions, and therefore the available data is limited.

While it is possible to collect the TIS data for the winter from 2022 to 2023, it is expected that there will be differences in detection performance between seasons. The aim of this study is to demonstrate the feasibility of the SDGSAT-1 TIS, a new 30 resolution sensor, for lead detection. Thus, despite the limited data, our results suggests that the TIS is competitive and promising for application in Arctic lead observations.

Currently, SDGSAT-1 needs to take into account various imaging requirements in different areas, so it is difficult for the satellite to keep observing the polar regions for long periods. The SDGSAT-1 data over the Arctic are still ongoing collected. Besides, the three payloads of SDGSAT-1 (TIS, Glimmer Imager for Urbanization (GIU) and Multispectral Imager for Inshore (MII)) allow for daytime and nighttime atmosphere monitoring capabilities. The corresponding SDGSAT-1 cloud product is under development.

The TIS lead map is evaluated against the Sentinel-2 lead map, and their agreement is very high, 96.3%. You talk about accuracy of the TIS lead map based on this comparison, but this comparison really don't give an absolute accuracy of the TIS map, as I don't think your Sentinel-2 lead map is error free. For determination of accuracy in-situ or airborne data are needed.

**Response:** We agreed with you that it would be more valuable to compare in-situ and airborne measurement. For example, the recent 1 m resolution IST data based on flight-borne thermal infrared camera in the MOSAIC expedition is interesting (Thielke et al., 2022), while this dataset only covers the central Arctic. Field data is, after all, scarce and hard to match with our SDGSDA-1 TIS data, and supposedly are well beyond the scope of this study.

In terms of validation of the accuracy of lead detection, previous studies based on moderate resolution thermal infrared remote sensing have also used a variety of different methods. For example, Willmes and Heinemann (2015) used the normalized brightness images derived from MODIS near-infrared (NIR) data (channel 2, 841–876 nm) with a resolution of 250m × 250m for validation. Hoffman et al. (2019) compared their time-series results with the Willmes and Heinemann (2015). Hoffman et al. (2021) used the masks derived from hand analysis as a proxy for validation in the absence of ground truth in the Arctic and conducted a comparison with the legacy product by Hoffman et al. (2019). Qu et al. (2021) used three successive Landsat-8 NIR images at 30 m resolution to assess the accuracy of their lead detection. As for other lead detection studies, Murashkin and Spreen (2018) evaluated the S1-derived lead results in comparisons with S2 optical satellite data. But instead of assigning reflectance thresholds to the S2 data, they manually marked the S1 data by overlaying the S2 image to confirm the validity of leads.

Overall, even with certain errors, it is sound to use S2 data with normalized brightness and objective companions for validation in this study.

TIS strip noise is discussed first time under Results. It should be mentioned when TIS sensor and data properties are described under Section 2.

It would be very interesting see lead map comparison between TIS (30 m) and Landsat (100 m), can you add this to your paper? We could see how much 30 vs. 100 m resolution matters in the lead detection.

Finally, under 'Summary and conclusion" Section you discuss more about future research goals, and will be there an operational TIS lead product?

**Response:** Landsat-8 at 100 m resolution is indeed appropriate to be used for comparison with the TIS results. However, we did not acquire the matched Landsat-8 data during SDGSAT-1 TIS imaging. So, we only compared with the MODIS data at 1km resolution. The figure below shows the coverage of Landsat-8 in the Beaufort Sea on 3 April 2022 (available on the United States Geological Survey website,

[Figure]

Figure R1 The search result for Landsat-8 in the Beaufort Sea on 3 April 2022 (https://www.usgs.gov/)

In the future, we do plan to develop a long-term lead dataset based on SDGSAT-1 TIS at 30 m resolution to support relevant research about sea ice dynamic, which requires more SDGSAT-1 data accumulation and development of related products (particularly cloud products and surface temperature products).

**Specific comments**

**Abstract**

"unresolvable ice leads"; change to "unresolvable sea ice leads" or to "unresolvable leads"

**Response:** Done as it is suggested.

**1. Introduction**

line 28: "under wind and water stresses" better "under wind and ocean stresses".

**Response:** Done as it is suggested.

l. 33: "During winter, newly opened leads are the main source of ice production, brine rejection, and turbulent heat loss to the atmosphere"

Is it leads or polynyas in the whole Arctic scale? Please check. Anyway polynyas could be mentioned in this context.

**Response:** Done as it is suggested. The sentence is modified "During winter,

newly opened leads and polynyas are the main source of ice production, brine rejection, and turbulent heat loss to the atmosphere"

l. 47: "Other studies also applied active and passive microwave data to lead detection with the advantage that microwave wavelengths are transparent to cloud cover; however, either the data resolution is too coarse"

Too coarse to what? Detect smaller leads? In some application of the lead data?

**Response:** The "coarse resolution" here meant that the leads derived from AMSR-E with a resolution of 6.25 km is coarser than the moderate resolution thermal infrared (with resolutions of hundreds of or thousands of meters) and high-resolution optical images (tens of meters of resolution). Some studies shown that the 6.25 km resolution lead dataset is not ideal for the evaluation of narrow leads. Ólason et al. (2021) used the AMSR-E lead dataset developed by Bröhan and Kaleschke (2012a) to evaluate the ability of a new sea ice model in reproducing lead characteristics. They found that "small leads are known not to be captured by the AMSR-E because of its resolution limitation". Including, but not limited to, Ivanova et al. (2016) applied SAR data to assess the error in the AMSR-E lead dataset and found a consistent overestimation of lead fraction by a factor of 2 to 4 in the AMSR-E product.

Ivanova, N., Rampal, P., and Bouillon, S.: Error assessment of satellite-derived lead fraction in the Arctic, The Cryosphere, 10, 585–595, https://doi.org/10.5194/tc-10-585-2016, 2016.

l. 57: "Essentially, IST data, which are generally retrieved by the split-window technique (Key et al., 1997), are less accurate under the presence of melt ponds and leads because the lower emissivity (0.96 compared to 0.99) can cause a difference in the retrieved temperature"

Lower emissivity of what? Water?

**Response:** Since melt ponds and leads contain water, they have lower emissivity than that of snow or ice, say 0.96 compared to 0.99. Please refer to Key et al. (1997) and Hall et al. (2001) for details. The sentence is revised "because the lower emissivity (0.96 compared to 0.99) of water than sea ice".

l. 61: "They detected leads for January through April over the period of 2003 to 2018, presenting a lower estimation for the lead area compared with the results in Willmes and Heinemann (2015c); the reason is the difference in spatial resolutions of the lead datasets."

Give resolutions of these datasets in the text.

**Response:** We have modified this to "the reason is the difference in spatial resolutions of the lead datasets (1 km compared to 2 km)".

l. 67: "Qu et al. (2021) proposed a modified algorithm Modified from what? From Hoffman et al?

**Response:** The sentence is revised "Qu et al. (2021) proposed a modified algorithm from Willmes and Heinemann (2015a) to detect daily spring leads in the Beaufort Sea based on the IST data retrieved from MODIS swath products".

l. 94: "To date, the TIS has acquired substantial high-resolution thermal infrared data from the critical seas in the Arctic"

How do you define what is a critical area in the Arctic? Explain in the text.

**Response:** In the scope of this study, the critical seas refer to areas pervaded by leads with significant sea ice dynamic process.

Wernecke and Kaleschke (2015) showed that the lead fraction in Baffin Bay, the Fram Strait region, the northern Barents Sea, the Kara Sea, the western Laptev Sea and the Chukchi Sea are commonly up to around 15 %. In the southern Beaufort Sea and especially its shear zone next to the coastline, lead fraction values are up to 6 %. A strong lead divergence and opening processes were observed in the Beaufort Sea (Beitsch et al., 2014). The model simulation by Wang et al. (2016) also suggested that winter ice leads are mainly formed in marginal seas (Barents, Kara, Laptev, and Beaufort Seas) and near Fram Strait. In particular, leads fraction in the Beaufort Sea shows a significant interannual variability.

Thus, the Beaufort Sea and Laptev Sea can be considered to as the critical areas for lead observation.

Beitsch, A., Kaleschke, L., and Kern, S.: Investigating high-resolution AMSR2 sea ice concentrations during the February 2013 fracture event in the Beaufort Sea, Remote Sensing, 6, 3841-3856, 2014.

Wang, Q., Danilov, S., Jung, T., Kaleschke, L., and Wernecke, A.: Sea ice leads in the Arctic Ocean: Model assessment, interannual variability and trends, Geophysical Research Letters, 43, 7019-7027, 2016.

l. 102: "This study is the first to observe Arctic sea ice leads at 30 m resolution"

**Response:** This has been modified to "This study focuses on observing Arctic sea ice leads based on spaceborne thermal infrared remote sensing at 30 m resolution and reveals more details than the moderate-resolution thermal infrared sensors."

Table 1 must include references to the data it present.

**Response:** Agree. The reference is:

> International Research Center of Big Data for Sustainable Development Goals (CBAS): SDGSAT-1 Data Users Handbook (Draft), 2022. http://60.245.209.56/preview/20221125/c84c 0b5d89984cd384ffa05dbb163d14.pdf

Figure 1 should include acquisition times for the rectangles 1-4. Describe that visible images come from Sentinel-2.

**Response:** We have added corresponding dates and descriptions to the notes of Figure 1: "The black borders mark four successive groups of cloudless images (group 1 was acquired on 3 April, group 2 on 28 April, groups 3 and 4 on 23 March)". Please refer to the Table 2 in the original manuscript for more details of data information.

**2. Data**

l. 130: "Considering the benefit of incorporating three thermal infrared bands for observation, Thus, the three bands of SDGSAT- 1 TIS data are used"

You can remove 'thus' from the sentence.

**Response:** Done as it is suggested.

l. 131: "The georeferenced level-4 TIS data"

What do you mean by level-4 data here?

**Response:** The SDGSAT-1 data products include different Level-1, Level-2 and Level-4 data products. Level-1 product is a standard product based on the Level-0 product, after data processing such as relative radiometric correction, band registration, HDR fusion, etc. Level-2 product is based on the Level-1 product after geometric correction. Level-4 product is based on the Level-1 product after ortho-rectification using ground control points and Digital Elevation Model (DEM) and output with standardized format. Currently, only Level-4 product is available to

users (CBAS, 2022).

Give some refences to Sentinel- 1 and -2 sensors and data.

**Response:** Done as it is suggested. The User Guides for S1 and S2 are cited as references:

Eurpean Space Agency (ESA): Sentinel-1 User Handbook, 2013.

Eurpean Space Agency (ESA): Sentinel-2 User Handbook, 2015.

l. 139: "given that the visible spectrum centered at 560 nm gives a good effect (König et al., 2019) for a scene containing sea ice and seawater. "

What is this 'effect'? Good discrimination between sea ice and water?

**Response:** We visually compared S2 visible images (bands 2, 3 and 4) and found good discrimination between leads and surrounding sea ice from the band 3. Therefore, it is appropriate to use the S2 green band images as the reference for validation.

l. 147: "we collected S1A dual-polarization data in the Beaufort Sea on April 3 and 28, 2022 (see Table 2).

Table 2 shows S- 1 data only on 3 Apr.

**Response:** We have corrected this in the revised version. It should be "we collected S1A dual-polarization data in the Beaufort Sea on April 3 in 2022".

Table 2: S-1 resolution is not 40 m, it is around 100 m, the pixel size is 40 m. Explain what are h07/08 etc. under MODIS column.

**Response:** Thanks for pointing these out. We have changed the information about S1 data. It should be "pixel size: 40 m" for the Sentinel-1 EW data.

For the level-3 MOD29 product, "hxx" and "vxx" are the horizontal tile number and vertical tile number of product. We have taken on the advice of reviewer 1 and conducted the experiment using the level-2 MOD29 product. The level-2 MOD29 data information are listed in Table R3.

Table R3 Information of MODIS products used in this study

| | | MOD29 | MOD03 |
|---|---|---|---|
| Date and time (UTC) | 2022-03-23 | 10:30 12:05 | 10:30 12:05 |
| | 2022-04-03 | 05:10 | 05:10 |
| | 2022-04-28 | 05:05 | 05:05 |
| Spatial resolution | | 1 km | 1 km |

l. 163: "air temperature) data available by every 6 hours" Your ERA5 reference shows that data is hourly.

**Response:** We have corrected this. It should be "The ERA5 near-surface air temperature (2 m air temperature) data is available hourly in a regular grid of 0.25 degrees."

OMI ozone: give some reference also for the ozone retrieval, in addition to the product reference.

**Response:** Agree. The literature about ozone retrieval is cited, with the corresponding description: "The total column ozone is retrieved based on the long-standing TOMS V8 retrieval algorithm (Bhartia, 2002), which uses a weakly absorbing wavelength (331.2 nm) to estimate an effective surface reflectivity and another wavelength (317.5 nm) with stronger ozone absorption to estimate ozone.".

Bhartia, P. K.: OMI Algorithm Theoretical Basis Document, Volume II, OMI Ozone Products, Greenbelt, Maryland, USA, NASA Goddard Space Flight Center2002.

**3. Method**

l. 195: "In addition to the leads presenting as distinct yellow and red colors on the BT map, Give also temperature ranges these colors represent.

**Response:** The sentence was modified to "the leads presenting as distinct yellow and red (in the temperature range of 242 K to 252 K) colors on the BT map".

Give some references to the BTA based lead detection in Section 3.2, as this method has been used in many studies using MODIS and VIIRS data.

**Response:** Agree. We added relevant references at the beginning of Section 3.2.

"Ice leads containing seawater and thin ice have temperatures higher than the surrounding sea ice. Therefore, the temperature contrast between leads and the sea ice surface is the basis of detecting ice leads (Willmes and Heinemann, 2015a; Hoffman et al., 2019; Qu et al., 2021)."

l. 208: "By collecting eight TIS data acquired between April 3 and April 28" There is seven TIS images in Table 2.

**Response:** Thanks for pointing it out. It should be seven TIS images. We have corrected this in the manuscript.

l. 222: "Previous methods applied a variety of BTA thresholds"

should be "Previous studies", and give also references.

**Response:** Done. The sentences have been modified "Previous methods applied a variety of BTA thresholds. Hoffman et al. (2019) identified a threshold of 1.5 K. Qu et al. (2021) took 1.2 K, 1.5 K and 2 K as thresholds for different types of leads, corresponding to large to small uncertainty levels.".

Figure 6 and 7: give TIS band used in the figures.

**Response:** Done as it is suggested. The used SDGSAT-1 TIS data ID (KX10_TIS_20220403_W128.84_N73.00_202200033226) is the same as in Figures 4 and 5.

l. 268: "Finally, the binary detection of leads at a 30 m resolution was derived based on SDGSAT- 1 TIS in three bands."

How all three bands were used together in the lead detection? You must explain this in detail.

**Response:** The TIS data from each of the three bands was fed separately into the detection algorithm, and the outcomes were three binary lead maps. Except for Section 4.3, where we explicitly said that the three results were combined into a lead map, all the rest was based on three results from the three bands. In terms of the detection performance, each band has its advantage on leads detection, so we consequently suggest using the combined results of the leads detected from the three TIS bands.

The sentence is modified "Finally, three binary results at 30 m resolution were derived separately from each of the three bands of the SDGSAT-1 TIS.".

**4. Results**

l. 302: spell out TP, FP, FN and TN in the text,

Done

Section 4.2: equation numbers should be (2) and (3).

Done

Section 4.3: again how lead detections by all three TIS bands are combined?

**Response:** The sentence was modified: "Simultaneously, we combined the three lead maps based on the three TIS bands into one binary map, in which the combined pixel is positive as long as one of the three maps gives a positive pixel. The combined map contains the most leads (see Figure 11 (c))."

l. 381: "The previously developed method (Qu et al., 2021) was applied to detect the leads based on the MODIS IST data."

Explain in the text why you selected the method by (Qu et al., 2021).

**Response:** We used the method by Qu et al. (2021) to detect leads from MODIS IST data because it is based on an analogous principle to our proposed method, i.e., both used fixed BTA thresholds for binary segmentation. The use of analogous methods allows for a fair comparison of the differences in lead observations between the two sensors. We have added this to the revised version.

**5. Discussion**

You talk in the text about 'ozone resolution', what is that? Figure 12 shows 'best total ozone solution'.

**Response:** It is a typo. It should be the "ozone solution". We have corrected them.

You could add to the discussion here the correlations between different TIS bands.

**Response:** We have shown cross comparisons of the lead detection results from the TIS three band in Section 4.2. The three bands give highly consistent detection results. The inconsistent lead pixels detected from the three bands account for 11.46%, 23.30%, and 21.88%, respectively. The correlation between the thermal characteristics of the leads for the three bands is evident. Therefore, we did not further discuss this.

Section 5.2: Descriptions on S- 1 SAR processing should go to Data Section.

**Response:** Done as it is suggested. The SAR data processing has been amended in Section 2.2: "The dual-polarization data were radiometrically calibrated, and a false-color composition was performed by assigning the HH, the subtraction of HH by HV and the HV images to the red, green and blue channels, respectively."

l. 483: "In particular, the B2 band is more sensitive to such surface information because various types of sea ice have different emissivity and produce different BT values. "

Are the TIR emissivities of different sea ice types really that different? Please check literature, and give variation range in the text. I would say it is more about variation of sea ice/snow surface temperature, e.g. thin sea ice has surface temperature which is a function of freezing temperature and air temperature, but surface of thick ice is fully insulated from ice bottom, i. e. only air temperature matters.

**Response:** Thanks very much for your valuable comment. We have reviewed the relevant literatures. Indeed, as you say, TIR (thermal infrared) emitted energy mainly reflects the skin radiometric information of the ice/ snow. It is a function of surface temperature and emissivity, also influenced by meteorological conditions.

For TIR wavelength region, water emissivity is stable, about 0.96. Snow and ice have high emissivity, but with variations (Sandven and Johannessen, 2006). Theoretically, the emissivity of ice/snow decreases with increasing grain size and increasing density, enhancing the emissivity spectral contrast (Warren, 2019). Ice/snow type dependent emissivity can produce a brightness temperature difference. Laboratory measurements of terrestrial material emissivities indicate that snow/ice emissivity changes spectrally with snow/ice conditions such as grain size and packing fraction (Tonooka and Watanabe, 2005). The field measurement shows that the

emissivity of fine-grained snow is very high for the nadir angle, 0.997 and 0.984 for 10.5 μm and 12.5 μm. The emissivities of coarse-grained snow are 0.995 and 0.971, and become sensitive to viewing angle. The emissivity of bare ice is the lowest, 0.993 and 0.949 for 10.5 μm and 12.5 μm, respectively, with the largest angular dependence (Hori et al., 2006; Hori et al., 2013). Additionally, due to the mixed pixel effect in TIR remote sensing, the presence of unresolved melt ponds and leads can also change the surface emissivity (Hall et al., 2001).

Nevertheless, the differences in the ice/snow surface emissivity in the TIR spectra above mentioned are small. The effect of mixed pixels is also small, especially for the fine scales of focus in this study. In contrast, ice/snow thickness can vary significantly over short distances and on ice of different ages. The morphology of the snow cover, in conjunction with the air temperature near the surface, can be expected to be a significant cause of surface temperature variations (Poulin, 1975).

Hence, as you say, the variations in BT values for the SDGSAT-1 TIS in this study is more likely to be caused by ice/snow surface temperatures. For thin ice, its surface temperature is close to the temperature of surrounding icefree water and underside water depending on the thickness. For thick ice, it is covered by snow and insulated from ice bottom, so its surface temperature is generally close to that of surface air temperature (Comiso et al., 2019).

Thielke et al. (2022) obtained sea ice surface temperatures at 1 m resolution based on helicopter-borne thermal infrared images in the MOSAiC expedition during winter 2019/2020. The high-resolution temperature images show significant warmer line features, i.e., opened leads, with internal surface temperature variability. They also found the high variability of surface temperature for surrounding sea ice at high resolution, and explained it as the result of difference ice thicknesses. This temperature variation characteristic is similar to the situation we described in our Figure 13 and therefore supports our analysis.

The corresponding paragraph has been amended: "contours of multiyear ice with

high backscatter values that are observed in SAR images are similar to some negative BTA features... This suggests surface temperature variations for different thicknesses of sea ice. Similar surface temperature variations are also found in the 1 m resolution IST data derived from helicopter-borne thermal infrared imaging (Thielke et al., 2022).".

Poulin, A. O.: Significance of surface temperature in the thermal infrared sensing of sea and lake ice, Journal of Glaciology, 15, 277-283, 1975.

Sandven, S. and Johannessen, O. M.: Sea ice monitoring by remote sensing, 2006

Hori, M., Aoki, T., Tanikawa, T., Motoyoshi, H., Hachikubo, A., Sugiura, K., Yasunari, T. J., Eide, H., Storvold, R., and Nakajima, Y.: In-situ measured spectral directional emissivity of snow and ice in the 8–14 μm atmospheric window, Remote Sensing of Environment, 100, 486-502, 2006.

Hori, M., Aoki, T., Tanikawa, T., Hachikubo, A., Sugiura, K., Kuchiki, K., and Niwano, M.: Modeling angular-dependent spectral emissivity of snow and ice in the thermal infrared atmospheric window, Applied optics, 52, 7243-7255, 2013.

Tonooka, H. and Watanabe, A.: Applicability of thermal infrared surface emissivity ratio for snow/ice monitoring, Multispectral and Hyperspectral Remote Sensing Instruments and Applications II, 282-290, 2005

Warren, S. G.: Optical properties of ice and snow, Philosophical Transactions of the Royal Society A, 377, 20180161, 2019.

Josefino C. Comiso, Dorothy K. Hall, and Rigor, I.: Ice Surface Temperatures in the Arctic Region, in: Taking the Temperature of the Earth, edited by: Glynn C. Hulley, and Ghent, D., Elsevier, 10.1016/B978-0-12-814458-9.00005-8., 2019.

Thielke, L., Huntemann, M., Hendricks, S., Jutila, A., Ricker, R., and Spreen, G.: Sea ice surface temperatures from helicopter-borne thermal infrared imaging during the MOSAiC expedition, Scientific Data, 9, 364, 10.1038/s41597-022-01461-9, 2022.

l. 498: "On the other hand, as the TIS data available within the scope of this paper is relatively limited, these individual case studies presented may be weak in terms of generalizability."

Yes, this is the case, and this must be also emphasized in Conclusion Section.

**Response:** We agree with you about the limitations of this study. We have mentioned it in the Conclusion Section: "Nevertheless, limited by the imaging time and cloudy conditions over the Arctic region, only individual case studies based on TIS data were carried out.".

**Technical corrections**

Many figures are too small, try to increase their sizes. Also colorbars for BT, BTA, are way too small.

References for the same authors and the same year are missing a,b,c, after the year in Reference list.

**Response:** Thank you very much for your detailed advice. We will improve the quality of these figures and check the citation format of the literatures.

---

## Referee Report (RR1)

**Spaceborne thermal infrared observations of Arctic sea ice leads at 30 m resolution – revision 1**

Yujia Qiu, Xiao-Ming Li, Huadong Guo

**8 May 2023**

The authors have presented detailed, proper answers to most of my comments to the original version of the paper, and made corresponding changes and additions to the paper. I think that the paper has improved considerably. Below I have some comments for your consideration for further paper improvements.

From first review: "I think the study set up with data acquisitions and data processing is rather well conducted, and the TIS lead detection method could be generally applicable for a large number of TIS images, but I think it is not sure as it is based on small amount of data. Further, I don't think it is meaningful to develop lead detection methods for the three TIS bands separately, and compare the results. You should develop the best possible lead detection algorithm for the TIS data (having as input all bands or just two/one), and only present this in the paper. In the following I have also some other major comments to the papers and suggestions for possible improvements. These are followed by miscellaneous specific ones."

*Response: Thanks for your valuable comment. We agree that it would be better to apply the most appropriate thermal infrared band for lead detection. However, until the cross-comparison experiments were carried out, we found each band has its advantage on ice leads detection.*

*Taken together, it is necessary to carry out a comprehensive application demonstration of the new on-board sensor. We applied the same method to the three bands of SDGSAT-1 TIS to extract leads, and further conducted cross-comparison to determine their detection performance. The cross-comparison results suggest it is beneficial to combine the lead detection results of the TIS three bands.*

> I can understand your view here, and you compared lead maps by the three bands and also combine them together in Section 4.3 for comparison the MODIS maps, but it still would be nice see a case study where all three bands, or at least two bands, are used together for the lead detection, and whether this brings any improvements.

*The absolute radiometric calibration evaluation by Hu et al. (2022) suggested that the average temperature bias of SDGSAT-1 TIS reached 0.661 K, 1.081 K and 0.426 K for B1, B2 and B3, respectively. This suggests that the B3 band has the best radiometric calibration accuracy. However, B2 and B3 bands are more affected by the strip noise than the B1 band. B1 band is a less common thermal infrared channel with colder temperatures than the other two split-window channels (B2 and B3 bands) due to the absorption effect of ozone However, we do find that using the TIS B1 band can obtain more small leads in the presence of interference in B2 and B3 data.*

> Not all this information is in revised Section 2.1, e.g. band-wise accuracy information.

Table 1 on currently available lead detection datasets is very good addition to the paper.

You have added some references on previous lead studies which I suggested to the paper, and omitted those which were not relevant. I agree with your decisions.

On automated SAR lead detection:

*Therefore, the automated lead detection algorithm may not be adapted to the scenario we have shown here. In contrast, it is more appropriate to analyze the differences between HH and HV data directly, so we have performed a false-color composite using the dual-polarized data.*

Yes, I can agree with this. You could explain in the paper why automatic methods were not applied.

*Response: In accordance with your suggestions, we amended the data presentation and pre-processing sections (please refer to the revised version if the editor decides the manuscript can be revised). We would like to answer your questions about the TIS data here (and have added these details where appropriate in the revised manuscript).*

I think Section 2 is now much better with more information on the TIS data.

*For convenient use of the TIS data, the ground segment crops the original TIS data to 300 km in the along-track dimension.*

*TIS B1 band is a wide channel with a wavelength of 8.0-10.5 μm. It is mainly used in combination with the B2 (10.3-11.3 μm) and B3 (11.5-12.5 μm) bands to obtain a better accuracy in land surface temperature retrieval based on the three-channel split-window algorithm (Liu et al., 2021; Hu et al., 2022).*

*The quantization bit of the TIS is 12 bit. The TIS radiometric measurement is better than 0.42 K for the three bands, which satisfies the preflight requirements (≤1 K)*

Please add these to Section 2.1.

*There are currently no TIS-based surface temperature products or cloud mask products available, all of which are under development.*

This is very important information and must be added to Section 2.1.

On the automatic BTA threshold determination:

*Response: We did consider using iterative thresholds for the BTA data as well, as Willmes and Heinemann (2015a) did. However, it is hard to argue that automatically selected thresholds are more appropriate than fixed thresholds for few cases in this study. For the three bands lead detection, without the use of a fixed BTA threshold for standard, the comparability of binary segmentation results would be poor, and the further cross-comparisons between the three results would be meaningless.*

Yes, you are right here, for your study the use of manually determined BTA threshold is ok.

*Although the TIS data used in this study cannot yet include various sea ice and atmospheric conditions, we would like to explain here the soundness of the constant threshold. We tested the results of the threshold values selected by the iterative method. Setting the initial threshold as 1.2 K, the automatically selected BTA thresholds by iterative method for the seven TIS data (for the each of the three bands) were shown in Table R2. The iteration thresholds for the BTA images were relatively close, with the minimum of 1.8 K and the mean of 2.0 K. From this perspective, no large errors can be produced between the segmentation results from iterative selection or from the constant threshold.*

You could add to the paper a short mention about this automatic BTA threshold determination study, and that the results were close to the manual BTA threshold.

On adding more TIS data:

*Response: Although we would like to carry out more detections, what we have presented in this manuscript is all that can be done in the spring of 2022. On the one hand, the SDGSAT-1 was launched just one year ago, so we can only obtain data after March 2022. On the other hand, the cloud interference is the main limitation for lead detection based on thermal infrared data in the Arctic. Due to the unavailability of simultaneous cloud detection (we are also working on this*

*point), the method proposed in this study is only concerned with clear sky conditions, and therefore the available data is limited.*

OK, you could add launch date of SDGSAT-1 to Section 2.1 (it is now in Section 1, but could be repeated), and that practically you have all available data used in your paper at the time of its writing.

*Currently, SDGSAT-1 needs to take into account various imaging requirements in different areas, so it is difficult for the satellite to keep observing the polar regions for long periods.*

Please add this to Section 2.1.

On the comparison of the TIS lead map against Sentinel-2 lead map:

*Response: We agreed with you that it would be more valuable to compare in-situ and airborne measurement.*

*In terms of validation of the accuracy of lead detection, previous studies based on moderate resolution thermal infrared remote sensing have also used a variety of different methods.*

*Overall, even with certain errors, it is sound to use S2 data with normalized brightness and objective companions for validation in this study.*

Yes, it is ok to use S2 lead maps as validation data, but you should take care of using the results to show accuracy of the TIS lead map, as they more show how two remote sensing products agree. It does not to tell what is the absolute accuracy of your TIS lead map, to my opinion.

On Landsat vs. TIS data:

*Response: Landsat-8 at 100 m resolution is indeed appropriate to be used for comparison with the TIS results. However, we did not acquire the matched Landsat-8 data during SDGSAT-1 TIS imaging. So, So, we only compared with the MODIS data at 1km resolution.*

Yes, I understand, and comparison to Landsat can be left for future studies.

*In the future, we do plan to develop a long-term lead dataset based on SDGSAT-1 TIS at 30 m resolution to support relevant research about sea ice dynamic, which requires more SDGSAT-1 data accumulation and development of related products (particularly cloud products and surface temperature products).*

Please add this to Section 6.

I don't have any new major comments to the revised paper.

**Specific comments**

The authors have resolved nicely my specific comments, below are few comments to their responses:

How do you define what is a critical area in the Arctic? Explain in the text.

*Response: In the scope of this study, the critical seas refer to areas pervaded by leads with significant sea ice dynamic process.*

*Thus, the Beaufort Sea and Laptev Sea can be considered to as the critical areas for lead observation*

Please add this explanation to the paper.

2. Data

l. 131: "The georeferenced level-4 TIS data"

*Level-4 product is based on the Level-1 product after ortho-rectification using ground control points and Digital Elevation Model (DEM) and output with standardized format.*

Add this level-4 data description to Section 2.1.

5. Discussion

l. 483: "In particular, the B2 band is more sensitive to such surface information because various types of sea ice have different emissivity and produce different BT values."

*The corresponding paragraph has been amended: "contours of multiyear ice with high backscatter values that are observed in SAR images are similar to some negative BTA features... This suggests surface temperature variations for different thicknesses of sea ice. Similar surface temperature variations are also found in the 1 m resolution IST data derived from helicopter-borne thermal infrared imaging (Thielke et al., 2022).".*

This is very good correction and clarification to the paper.

Related discussion in Section 1, line 60:

"Essentially, IST data, which are usually retrieved using the split-window technique (Key et al., 1997), are less accurate in the presence of melt ponds and leads because of the lower emissivity (0.96 compared to 0.99) of water compared to sea ice, causing a difference in the retrieved temperature (Hall et al., 2001)."

I don't quite understand this, why IST has less accuracy in melting conditions (melt ponds present)? Does emissivity difference 0.96 vs. 0.99 matter that much? In melting, i.e. warm conditions, we have very little, if any, thermal (IST) contrast between different surface types, is this what you are really meaning?

l. 498: "On the other hand, as the TIS data available within the scope of this paper is relatively limited, these individual case studies presented may be weak in terms of generalizability."

Yes, this is the case, and this must be also emphasized in Conclusion Section.

---

## Author Response (AR2)

**Response to Review**

Dear Editors and Reviewers:

We would like to thank the reviewer for the detailed and helpful comments, suggestions, and careful checking. We have addressed the comments on a point-by-point basis and implemented the proposed changes in the manuscript. The comments from the reviewer are in black while our responses are highlighted in blue. We also provide a track-change version together with the revised paper.

The authors have presented detailed, proper answers to most of my comments to the original version of the paper, and made corresponding changes and additions to the paper. I think that the paper has improved considerably. Below I have some comments for your consideration for further paper improvements.

We appreciate the time and effort you dedicated to our manuscript. The enhancement of the paper would not have been possible without your valuable suggestions.

From first review: "I think the study set up with data acquisitions and data processing is rather well conducted, and the TIS lead detection method could be generally applicable for a large number of TIS images, but I think it is not sure as it is based on small amount of data. Further, I don't think it is meaningful to develop lead detection methods for the three TIS bands separately, and compare the results. You should develop the best possible lead detection algorithm for the TIS data (having as input all bands or just two/one), and only present this in the paper. In the following I have also some other major comments to the papers and suggestions for possible improvements. These are followed by miscellaneous specific ones."

*Response: Thanks for your valuable comment. We agree that it would be better to apply the most appropriate thermal infrared band for lead detection. However, until the cross-comparison experiments were carried out, we found each band has its advantage on ice leads detection.*

*Taken together, it is necessary to carry out a comprehensive application demonstration of the new on-board sensor. We applied the same method to the three bands of SDGSAT-1 TIS to extract leads, and further conducted cross-comparison to determine their detection performance. The cross- comparison results suggest it is beneficial to combine the lead detection results of the TIS three bands.*

I can understand your view here, and you compared lead maps by the three bands and also combine them together in Section 4.3 for comparison the MODIS maps, but it still would be nice see a case study where all three bands, or at least two bands, are used together for the lead detection, and whether this brings any improvements.

Thank you for your suggestion. The point you described is indeed very interesting. However, as the reviewer has mentioned, this study only presents a limited number of case studies, so it may not yet be sufficient to give very convincing results in terms of exploring how the three thermal infrared bands can (or should) be combined for lead detection. As a study applying new satellite data, our main concern here is whether the lead detection methods widely used by previous studies based on

thermal infrared data (Willmes and Heinemann, 2015a; Hoffman et al., 2019; Qu et al., 2021) are applicable to SDGSAT-1 TIS data. Our results show that while the temperature anomaly method can segment leads, additional step is needed to filter the sea ice surface temperature variations at high resolution.

As for the use of two or three bands for lead detection, in fact, we are currently exploring sea ice surface temperature retrieval methods based on SDGSAT-1 TIS, which involve the use of different thermal infrared bands such as split-window using the B2 and B3 or triple split-window using the B1, B2 and B3. This approach has the potential to lead to improvements in the further sea ice lead detection. However, regardless of the specific multi-band combination or subsequent lead detection, it would require the development of new methods (e.g., reconsidering appropriate temperature thresholds), which is beyond the scope of this study.

When more SDGSAT-1 data becomes available in the future, we will try to explore the interesting question you have raised. We mentioned this in the appropriate place in Section 6:

"We therefore recommend using the combined results of the leads detected from the three TIS bands and also intend to further explore the adaptability of combining different thermal infrared bands and their potential for improved lead detection in the future.".

*The absolute radiometric calibration evaluation by Hu et al. (2022) suggested that the average temperature bias of SDGSAT-1 TIS reached 0.661 K, 1.081 K and 0.426 K for B1, B2 and B3, respectively. This suggests that the B3 band has the best radiometric calibration accuracy. However, B2 and B3 bands are more affected by the strip noise than the B1 band. B1 band is a less common thermal infrared channel with colder temperatures than the other two split-window channels (B2 and B3 bands) due to the absorption effect of ozone However, we do find that using the TIS B1 band can obtain more small leads in the presence of interference in B2 and B3 data.*

Not all this information is in revised Section 2.1, e.g., band-wise accuracy

We have added the accuracy of the TIS three bands into Section 2.1:

"…the analysis shows that the accuracy of the radiometric measurement is better than 0.661 K, 1.081 K and 0.426 K for B1, B2 and B3 bands (Hu Y. et al., 2022)"

Table 1 on currently available lead detection datasets is very good addition to the paper.

You have added some references on previous lead studies which I suggested to the paper, and omitted those which were not relevant. I agree with your decisions.

Thanks for your comment.

On automated SAR lead detection:

*Therefore, the automated lead detection algorithm may not be adapted to the scenario we have shown here. In contrast, it is more appropriate to analyze the differences between HH and HV data directly, so we have performed a false-color composite using the dual-polarized data.*

Yes, I can agree with this. You could explain in the paper why automatic methods were not applied.

We have carefully clarified this point in Section 5.2.:

"Murashkin and Spreen (2018) developed an automated S1 lead detection algorithm. It should be noted, however, that this algorithm may have limited applicability for complex scenarios that involve a potential transition zone between thin ice and seawater. In contrast, the use of quantitative backscatter data obtained from dual-polarized S1 images has been found to offer improved

distinguishability."

*Response: In accordance with your suggestions, we amended the data presentation and pre-processing sections (please refer to the revised version if the editor decides the manuscript can be revised). We would like to answer your questions about the TIS data here (and have added these details where appropriate in the revised manuscript).*

I think Section 2 is now much better with more information on the TIS data.

The article has been enhanced thanks to the constructive comments provided by the reviewers. We thank the reviewers for their contributions.

*For convenient use of the TIS data, the ground segment crops the original TIS data to 300 km in the along-track dimension.*

*TIS B1 band is a wide channel with a wavelength of 8.0-10.5 μm. It is mainly used in combination with the B2 (10.3-11.3 μm) and B3 (11.5-12.5 μm) bands to obtain a better accuracy in land surface temperature retrieval based on the three-channel split-window algorithm (Liu et al., 2021; Hu et al., 2022).*

*The quantization bit of the TIS is 12 bit. The TIS radiometric measurement is better than 0.42 K for the three bands, which satisfies the preflight requirements (≤1 K)*

Please add these to Section 2.1.

Thanks for the comments. We have added these to the corresponding paragraphs in Section 2.2:

"…with a resolution of 30 m in a swath of 300 km (and the ground segment crops the original TIS data to 300 km in the along-track dimension for convenient use)"

"As a wide channel with a wavelength of 8.0-10.5 μm, the B1 band is commonly used in conjunction with the B2 and B3 bands with the aim of improving the precision of land surface temperature retrieval based on the three-channel split-window algorithm (Liu et al., 2021; Hu Z. et al., 2022)."

"Quantization bit: 12 bit" in Table 2 in the manuscript.

*There are currently no TIS-based surface temperature products or cloud mask products available, all of which are under development.*

This is very important information and must be added to Section

This has been added into Section 2.2: "Since the SDGSAT-1 was launched in November 2021, the development of TIS-based surface temperature products or cloud mask products is currently under development.".

On the automatic BTA threshold determination:

*Response: We did consider using iterative thresholds for the BTA data as well, as Willmes and Heinemann (2015a) did. However, it is hard to argue that automatically selected thresholds are more appropriate than fixed thresholds for few cases in this study. For the three bands lead detection, without the use of a fixed BTA threshold for standard, the comparability of binary segmentation results would be poor, and the further cross-comparisons between the three results would be meaningless.*

Yes, you are right here, for your study the use of manually determined BTA threshold is ok.

Thank you!

*Although the TIS data used in this study cannot yet include various sea ice and atmospheric conditions, we would like to explain here the soundness of the constant threshold. We tested the results of the threshold values selected by the iterative method. Setting the initial threshold as 1.2 K, the automatically selected BTA thresholds by iterative method for the seven TIS data (for the each of the three bands) were shown in Table R2. The iteration thresholds for the BTA images were relatively close, with the minimum of 1.8 K and the mean of 2.0 K. From this perspective, no large errors can be produced between the segmentation results from iterative selection or from the constant threshold.*

You could add to the paper a short mention about this automatic BTA threshold study, and that the results were close to the manual BTA.

Thanks for the comment. We have carefully added this into Section 3.1: "In addition to the fixed thresholds, we have also tested the iteratively selected thresholds (Willmes and Heinemann, 2015a) which yielded similar results to the manually selected fixed threshold of 1.8 K".

On adding more TIS data:

*Response: Although we would like to carry out more detections, what we have presented in this manuscript is all that can be done in the spring of 2022. On the one hand, the SDGSAT-1 was launched just one year ago, so we can only obtain data after March 2022. On the other hand, the cloud interference is the main limitation for lead detection based on thermal infrared data in the Arctic. Due to the unavailability of simultaneous cloud detection (we are also working on this point), the method proposed in this study is only concerned with clear sky conditions, and therefore the available data is limited.*

OK, you could add launch date of SDGSAT- 1 to Section 2.1 (it is now in Section 1, but could be repeated), and that practically you have all available data used in your paper at the time of its writing.

Thanks for the comment. We have clarified these in Section 2.1:

"Since the SDGSAT-1 was launched in November 2021, the development of TIS-based surface temperature products or cloud mask products is currently under development."

"The set of eleven TIS images, presented in Figure 1 and composed of four consecutive scenes, encompasses the majority of the available data up until the time of writing"

*Currently, SDGSAT-1 needs to take into account various imaging requirements in different areas, so it is difficult for the satellite to keep observing the polar regions for long periods.*

Please add this to Section 2.1.

Done as it suggested: "Considering diverse imaging requirements across various domains, it poses a challenge for SDGSAT-1 to maintain prolonged surveillance of polar regions.".

On the comparison of the TIS lead map against Sentinel-2 lead map:

*Response: We agreed with you that it would be more valuable to compare in-situ and airborne measurement.*

*In terms of validation of the accuracy of lead detection, previous studies based on moderate resolution thermal infrared remote sensing have also used a variety of different methods.*

*Overall, even with certain errors, it is sound to use S2 data with normalized brightness and objective companions for validation in this study.*

Yes, it is ok to use S2 lead maps as validation data, but you should take care of using the

results to show accuracy of the TIS lead map, as they more show how two remote sensing products agree. It does not to tell what is the absolute accuracy of your TIS lead map, to my opinion.

Yes, I have understood your point. The comparison between the TIS-detected and S2 lead results are not yet sufficient for the absolute accuracy of lead detection. We have revised the corresponding paragraphs in the manuscript, taking care to avoid any possible misleading reference to "overall accuracy". For example, in the abstract, we have revised:

"…the TIS-detected leads achieve good agreement with Sentinel-2 visible images."

On Landsat vs. TIS data:

*Response: Landsat-8 at 100 m resolution is indeed appropriate to be used for comparison with the TIS results. However, we did not acquire the matched Landsat-8 data during SDGSAT-1 TIS imaging. So, So, we only compared with the MODIS data at 1km resolution.*

Yes, I understand, and comparison to Landsat can be left for future studies.

Thank you!

*In the future, we do plan to develop a long-term lead dataset based on SDGSAT-1 TIS at 30 m resolution to support relevant research about sea ice dynamic, which requires more SDGSAT-1 data accumulation and development of related products (particularly cloud products and surface temperature products).*

Please add this to Section 6.

The corresponding sentences have been modified:

"Along with the acquisition of additional TIS data over the course of a year, as well as the development of a near-real-time cloud product, we plan to develop a long-term lead dataset based on SDGSAT-1 TIS at 30 m resolution to support research on the dynamics of sea ice and expect to investigate the lead detection capabilities of this dataset across different seasons."

I don't have any new major comments to the revised paper.

**Specific comments**

The authors have resolved nicely my specific comments, below are few comments to their responses:

How do you define what is a critical area in the Arctic? Explain in the text.

*Response: In the scope of this study, the critical seas refer to areas pervaded by leads with significant sea ice dynamic process.*

*Thus, the Beaufort Sea and Laptev Sea can be considered to as the critical areas for lead observation*

Please add this explanation to the paper.

The corresponding sentences have been modified:

"…the TIS has acquired substantial high-resolution thermal infrared data from the critical seas in the Arctic, including the Beaufort Sea and the Laptev Sea, which are pervaded by leads with significant sea ice dynamic processes (Wernecke and Kaleschke, 2015)."

2. Data

l. 131: "The georeferenced level-4 TIS data"

*Level‑4 product is based on the Level‑1 product after ortho‑rectification using ground control points and Digital Elevation Model (DEM) and output with standardized format.*

Add this level-4 data description to Section 2.1.

Done as it suggested:

"The georeferenced level-4 TIS data is based on the level-1 product after ortho-rectification using ground control points and Digital Elevation Model (DEM) and output with standardized format (CBAS, 2022).".

5. Discussion

l. 483: "In particular, the B2 band is more sensitive to such surface information because various types of sea ice have different emissivity and produce different BT values."

*The corresponding paragraph has been amended: "contours of multiyear ice with high backscatter values that are observed in SAR images are similar to some negative BTA features... This suggests surface temperature variations for different thicknesses of sea ice. Similar surface temperature variations are also found in the 1 m resolution IST data derived from helicopter‑borne thermal infrared imaging (Thielke et al., 2022).".*

This is very good correction and clarification to the paper.

Thanks for the comment.

Related discussion in Section 1, line 60:

"Essentially, IST data, which are usually retrieved using the split‑window technique (Key et al., 1997), are less accurate in the presence of melt ponds and leads because of the lower emissivity (0.96 compared to 0.99) of water compared to sea ice, causing a difference in the retrieved temperature (Hall et al., 2001)."

I don't quite understand this, why IST has less accuracy in melting conditions (melt ponds present)? Does emissivity difference 0.96 vs. 0.99 matter that much? In melting, i.e., warm conditions, we have very little, if any, thermal (IST) contrast between different surface types, is this what you are really meaning?

Thanks for pointing it out. Our statement here may have been misleading. It is not meant to imply limitations of the IST algorithm in melting scenes, but rather to highlight its challenges in scenes that involve both level ice and melting conditions (i.e., melting ponds and leads) due to the presence of mixed pixels.

In fact, this issue has already been mentioned in the "ATBD for the MODIS snow and sea ice-mapping algorithms" (Hall et al., 2001):

"The primary difficulty with surface temperature retrieval occurs when melt ponds and leads are present. The emissivity over water will be somewhat lower than that of snow or ice, say 0.96 compared to 0.99. This will make a difference of a few tenths of a degree (Jeff Key, written communication, 1996). The directional effects are also probably slightly different in melt ponds and leads as compared to snow- or ice-covered sea ice."

"…due to mixed pixel effects... the presence of melt ponds and leads in the summer months will also affect the emissivity of the ice surface and therefore the calculation of ice surface temperature."

We carefully revised the corresponding sentences in our manuscript as:

"Essentially, IST data, which are usually retrieved using the split-window technique (Key et al., 1997), has challenges in sea ice scenarios with the presence of melt ponds and leads. This is due to the lower emissivity (0.96 compared to 0.99) of water compared to sea ice, causing a difference in the retrieved temperature, especially with mixed pixel effects (Hall et al., 2001)."

l. 498: "On the other hand, as the TIS data available within the scope of this paper is relatively limited, these individual case studies presented may be weak in terms of generalizability."

Yes, this is the case, and this must be also emphasized in Conclusion Section.

Thanks for the comment. We had already emphasized this again in Section 6:

"Nevertheless, limited by the imaging time and cloudy conditions over the Arctic region, only individual case studies based on TIS data were carried out, which may result in a weak generalizability.".

---

## Author Response (AR3)

**Response**

Dear Editors and Reviewers:

We would like to thank our reviewers and editors for this opportunity. We gratefully acknowledge the helpful suggestions and careful checking of our manuscript. The content of this manuscript could only have been improved with their contributions.

For the specific technical suggestions, we will respond to them below and revise the corresponding context in the manuscript. A track-change version and the revised manuscript will be provided.

**Further suggestions for a minor technical revision:**

1. Which band of Sentinel-2 is shown in the figures? Probably band 3 (560 nm) as written in the text but this is not clear from the caption.

*__Response__: We used images from the Sentinel-2 band 3 (560 nm), as mentioned in Section 2.2. As suggested, we have also added this in the caption of Fig. 8.*

2. Not very relevant for this study but regarding the emissivity I wonder if there is a better citable source than the ATBD of Hall et al. (2001) which refers to Jeff Key, written communication, 1996?

*__Response__: Thanks for this suggestion. We will refer to other literature on the differences in IST retrieval due to surface emissivities in lead areas.*

*Fan et al. (2020) assessed the accuracy of different methods of ice surface temperature retrieval based on Landsat-8 thermal infrared imagery at 100 m resolution. They first discriminated between different types of surfaces, snow/ice surface and water surface (e.g., open water areas, ice-free leads, or flooded ice surfaces), and used different emissivities for them. Their results show that in the lead scene, the Landsat-8 retrieved surface temperatures are higher for lead areas than MODIS results, as the coarser resolution of MODIS tends to underestimate lead areas. This is essentially the same as an explanation by Hall et al. (2001) that the different emissivities due to the mixed pixel effect in thermal infrared remote sensing ultimately results in a difference in the retrieved temperature.*

*A more quantitative description of the result of this temperature difference can refer to Jiménez-Muñoz (et al., 2014). They note that part of the error in the surface temperature retrieval by the split-window algorithm arises partly from the uncertainty in the surface emissivity, which can contribute to a difference of 1.4 K.*

*Jiménez-Muñoz, J. C., Sobrino, J. A., Skoković, D., Mattar, C., and Cristobal, J.: Land*

*surface temperature retrieval methods from Landsat-8 thermal infrared sensor data, IEEE Geoscience and remote sensing letters, 11, 1840-1843, 2014.*

*We included this literature in the revised version to better clarify our idea: "Essentially, IST data, which are usually retrieved using the split-window technique (Key et al., 1997), has challenges in sea ice scenarios with the presence of melt ponds and leads. This is partly due to the lower emissivity (0.96 compared to 0.99) of water compared to sea ice, causing a difference in the retrieved temperature (Jiménez-Muñoz et al., 2014; Fan et al., 2020), especially with mixed pixel effects (Hall et al., 2001)."*

3. The link between improved detection of leads and the support of actual climate action (SDG 13) is in my opinion very weak. Maybe drop this half-sentence and the following appraisal of the satellite in the SDG context? It is certainly a great satellite mission, but the discussion of SDG indicators is beyond the scope of this journal. The sentence "Combining this data with diverse datasets of sea ice, we aim to provide insights into the contribution of leads to Arctic sea ice dynamics" would be a good closing.

*__Response__: Thanks for this suggestion. As suggested, we have removed the last paragraph. However, research related to Arctic warming and sea ice dynamics is also a significant concern of SDG13: climate action, which aims to "take urgent action to combat climate change and its impact," so we would like to retain at least this content. The manuscript will end with the following:*

*"Combining this data with diverse datasets of sea ice, we aim to provide insights into the contribution of sea ice leads to Arctic sea ice dynamics, in an effort to combat climate change and its impacts as support a key towards SDG 13: climate action.".*